# CHILDEVAL: HOW LARGE LANGUAGE MODELS MEET CHILDREN'S PERSONALITIES

## ABSTRACT

The remarkable success of Large Language Models (LLMs) has revolutionized LLM-based chatbots for personalized tasks beyond generic dialogues. Personalization involves customizing LLMs to generate text responses based on user preferences. One promising endeavor is to enable personalized interactions for children's caretakers while also promoting development and learning. However, dedicated research is required to determine whether LLMs can effectively deliver personalized responses based on children's preferences, given their interactions differ from adults and there are no child-oriented preference assessment methodologies. Facing these challenges, we introduce ChildEval, a benchmark to evaluate LLMs' capacity to infer, interpret, and follow child-centered preferences in a long-context conversational setting. Our benchmark comprises of 29K synthesized children's (ages 3-6) persona profiles, which are related to their preferences in both explicit and implicit manners. Implicit preferences are integrated inside dialogues consisting of 6 to 10 turns. The preferences cover 5 top-level and 14 sub-level topics that involve children's daily lives and development. We further propose fine-grained child-centric evaluation protocols to systematically evaluate the performance of open-source LLMs. Experimental results demonstrate the impact of various personalized representations on LLM responses and indicate that fine-tuning on this dataset may enhance performance. [1]

## 1 INTRODUCTION

Large Language Models (LLMs) (e.g., ChatGPT (OpenAI et al., 2024), Gemini (Gemini et al., 2025), and Claude (Anthropic, 2024)) have achieved remarkable success in effectively understanding and generating human language, leading to a revolutionary era in LLMs. Beyond generic dialogues, LLMs have been utilized in a wide range of individual daily tasks (e.g., healthcare (Xu et al., 2024) and finance (Easin et al., 2024)) to deliver personalized user experiences based on preferences (Kumar et al., 2024). One promising endeavor is to enable personalized interactions for children's caretakers while also promoting development and learning (Feng et al., 2024; Seo et al., 2024; Chen et al., 2025a), instead of just giving the "correct" answers.

Previous research into the potential of LLMs for personalized interactions has been focused on adult preferences and tasks. Qiu et al. (2025) advance LLMs toward effective personalization by introducing new methods for extracting user preferences from historical profiles. Other studies (Salemi et al., 2023; Jiang et al., 2025) attempt to address the challenge of a lack of adequate benchmarks for comprehensively evaluating the LLM's personalized capabilities. However, the proposed benchmarks focus on general preferences (e.g., the number of dialogue turns) with generic tasks (e.g., ticket booking and restaurant recommendations) for adults. There are several benchmarks (Rath et al., 2025; Liu & Fourtassi, 2024) proposed for children. Rath et al. (2025) examine child safety without considering a diversity of child-centered tasks. Liu & Fourtassi (2024) explore how LLM generate replies that mimic the child's style without considering the child's developmental and learning requirements. Dedicated research is required to determine whether LLMs can effectively deliver personalized responses based on children's preferences, as their interactions differ from those of adults.

---

[1] We will open-source all codes and data.

In particular, we identify the following primary gaps in current research on child personalization benchmarks in LLMs. 1) First, existing benchmarks fail to align with children's interactions, which show both non-conventional patterns and behaviors, e.g., word omissions, semantic errors, and incoherence (Liu & Fourtassi, 2024), while still responding in an age-appropriate manner (Chen et al., 2025b). 2) Second, there is a lack of a comprehensive taxonomy of evaluation protocols specific to children's personalization. Current personalized evaluation studies mainly cover standard adult-centered preferences (e.g., general preference following) and fail to address the specific needs of children (e.g., generating child-appropriate content while fostering children's creativity).

Facing these challenges, we introduce ChildEval, a benchmark to evaluate LLMs' capacity to infer, interpret, and follow child-centered preferences in a long-context conversational setting. We focus on the preschool children (aged 3-6) who have a greater need for companionship from LLM-based chabots. ChildEval comprises of 29K synthesized children's persona profiles, which are related to their preferences in both explicit and implicit manners. Implicit preferences are integrated inside dialogues consisting of 6 to 10 turns. The preferences cover 5 top-level and 14 sub-level topics that involve children's daily lives and development according to the guidelines published by the ministry of education (Antle, 2008; Wang, 2013). We further propose fine-grained evaluation protocols for child-oriented preferences to systematically evaluate the performance of open-source LLMs. Experimental results demonstrate the impact of various personalized representations on LLM responses and indicate that fine-tuning on this dataset may enhance performance.

## 2 RELATED WORK

### 2.1 DATA AND EVALUATION FOR PERSONALIZATION

Recent advances in LLM personalization have emphasized building data resources and evaluation benchmarks. PersonaMem (Jiang et al., 2025) and HiCUPID (Mok et al., 2025), for instance, simulate multi-attribute profiles and multi-turn conversations to assess whether models maintain user-specific consistency over time. Evaluation frameworks further consider metrics such as style alignment, preference fidelity, and user satisfaction (Salemi et al., 2023). On the data side, researchers have explored synthetic dialogue generation (Braga et al., 2024), profile summarization (Zhang, 2024), and memory retrieval from past interactions (Qian et al., 2025) to address the scarcity of high-quality personalized corpora.

Child-centered personalization has begun to emerge as a distinct line of research. KidLM (Nayeem & Rafiei, 2024) introduces a curated child-friendly corpus and training strategies for age-appropriate responses. Other works highlight the need for style simplification (Valentini et al., 2023) and safety evaluation frameworks tailored to children (Rath et al., 2025). Yet, most benchmarks and datasets target general users, leaving the open question of how well LLMs can adapt to children's preferences in multi-turn interactions.

### 2.2 METHODOLOGICAL ADVANCES IN LLM PERSONALIZATION

Methodological advances in personalization can be grouped into three main directions. First, *prompt-based methods* leverage explicit profile descriptions or inferred traits from past interactions. Structured attributes (e.g., demographics) improve personalization (Liu et al., 2025), while implicit profiles capture subtler user traits (Li et al., 2021), and profile placement affects model behavior (Wu et al., 2024). Second, *memory-based methods* incorporate past interactions as context, either directly or via retrieval systems such as MemPrompt (Madaan et al., 2022). Hierarchical memory structures (Pan et al., 2025; Magister et al., 2024) and integrated frameworks like PRIME (Zhang et al., 2025a) enable robust long-term personalization. Third, *preference modeling* seeks to capture user preferences. While LLMs still struggle with adherence (Zhao et al., 2025a), persona synthesis (Ryan et al., 2025) and representation editing (Zhang et al., 2025b) improve alignment.

Beyond non-parametric approaches, *parametric adaptation* embeds user traits into model parameters via supervised fine-tuning (full or parameter-efficient methods like LoRA or prefix-tuning). Representative works include OPPU (Tan et al., 2024), E2P (Huber et al., 2025), and controllable vectors such as Hydra (Zhuang et al., 2024). Reinforcement learning methods similarly adapt reward functions to individual preferences, e.g., PRLHF (Li et al., 2024) and RLPA (Zhao et al., 2025b).

While these methods benefit the general population, their effectiveness for children—characterized by implicit interests and markedly different communication patterns—requires new evaluation protocols.

## 3 THE CHILDEVAL BENCHMARK

### 3.1 PROBLEM FORMULATION

To evaluate whether an LLM can perceive and adapt to a child's preference $\rho$ when it communicates with the child, the full prompt sent to the model could be formulated by:

$$\mathcal{B} = H + u^* \tag{1}$$

where

- $+$ denotes the concatenation of texts.
- $H = \{S_1, S_2, \ldots, S_t, \ldots, S_T\}$ denotes a multi-session conversation history between a child and an LLM. Each session $S_t = \{(u_{t,1}, m_{t,1}), \ldots, (u_{t,K_t}, m_{t,K_t})\}$ consists of $K_t$ dialogue turns, where $u_{t,k_t}$ is the child utterance and $m_{t,k_t}$ the model response.
- $u^*$ is a child utterance related to the child preference $\rho$, and would be used as a utterance that the LLM shall respond to.

Sessions in $H$ are categorized as:

- **Relevant session**: Following the setting of Zhao et al. (2025a), the first session $S_1$ of $H$ is a session with dialogues related to the user preference queried by $u^*$.
- **Irrelevant session**: The remaining sessions of $H$ contain dialogue turns unrelated to $u^*$.

In each relevant session, the child preference $\rho$ can be revealed explicitly or implicitly:

- **Explicit**: Such a session contains a single dialogue turn $S_1 = \{(u_{1,1}, m_{1,1})\}$, and $u_{1,1}$ directly expresses the child preference.
- **Implicit**: Such a session contains multiple dialogue turns, and the user preference could be implicitly inferred by partial user dialogue turns in this session.

The task used for evaluating the LLM could then be formulated as:

$$f(p, \mathcal{B}) \longrightarrow \hat{m} \tag{2}$$

where $p$ denotes the child persona, i.e. the persistent attributes that embody the child's consistent personality traits (e.g., age and gender) and long-standing interests. $f(.)$ denotes the model to be evaluated, and $\hat{m}$ is the response generated by the model given the prompt $\mathcal{B}$. A good response shall align with the child persona $p$ and the child's preference $\rho$ revealed by the historical conversation displaced in $\mathcal{B}$.

Given such a problem formulation, a sample data of ChildEval is illustrated in Figure 1(b). The ChildEval benchmark comprises the child persona, the child preference statement, which is identical to the child utterance in the historical dialogue that explicitly reveals the preference, the historical multi-turn dialogues that implicitly reveal the preference, and a piece of preference-related child utterance used for LLM response evaluation. The composition of the benchmark enables us to assess not only a model's capability to identify and adapt to user preferences in dialogue but also how user personas influence the model's behavior patterns during interaction.

### 3.2 DATA CONSTRUCTION PIPELINE OF CHILDEVAL

Step 1: We generate 29K child personas using Qwen2.5-72B through an iterative generation-and-refinement process. Semantically similar candidates are removed using FAISS (Douze et al., 2024)

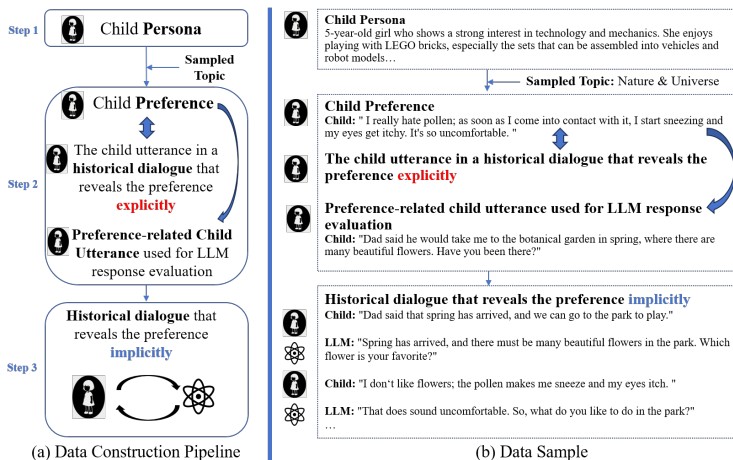

Figure 1: Overview of the ChildEval benchmark.(a) The data construction pipeline.(b) A data sample consists of the child persona, the child preference statement, which is identical to the child utterance in the historical dialogue that explicitly reveals the preference, the historical multi-turn dialogues that implicitly reveal the preference, and a piece of preference-related child utterance used for LLM response evaluation.

based on text embeddings (Xiao et al., 2023), ensuring diversity and comprehensive representation of personality traits. To mitigate privacy risks, all child names and identifiers are systematically removed via LLM processing and human review.

Step 2: Each child preference is generated from a child persona and a sampled topic, covering 5 top-level and 14 sub-level categories, following established guidelines on children's daily lives and development (Antle, 2008; Wang, 2013). For each child preference, a corresponding child utterance is also generated, which serves as the child's turn in a child-LLM dialogue, providing input for the LLM's response. Each child preference is expressed as a single first-person sentence and is linked to one sub-level topic and its corresponding top-level topic (see Table 1). Two preferences are generated per persona, yielding 58K preferences, of which 46K are retained after FAISS-based filtering of semantically similar instances.

Step 3: Historical dialogues implicitly revealing preferences are generated using prompt-based generation with self-verification, producing 6-10-turn child-LLM conversations. For more prompts, see Section A.5.

Table 1: Distribution of the 14 preference topics within ChildEval, which are related to children's daily life and development.

| Topic | Subtopic | | | |
|---|---|---|---|---|
| Art enlightenment (21.64%) | Music (6.68%) | Dance (6.74%) | Painting & Crafts (8.22%) | |
| Cognitive development and exploration (29.20%) | Science (7.72%) | Nature & universe (7.19%) | Technology (7.09%) | Learning (7.20%) |
| Nutrition and physical activity (13.11%) | Outdoor activity (6.24%) | Health eating (6.87%) | | |
| Language and literacy development (22.09%) | Story (7.39%) | Language (7.31%) | Reading (7.39%) | |
| Social and emotional development (13.97%) | Social interaction (6.94%) | Play (7.03%) | | |

## 3.3 FINE-GRAINED EVALUATION METRICS FOR CHILD PREFERENCES

As we stated previously, there is a lack of a comprehensive taxonomy of evaluation protocols specific to children's personalization. Existing personalized evaluation studies mainly cover standard adult-centered preferences. For example, Zhao et al. (2025a) and Jiang et al. (2025) focus on the

preference following performance from different settings. They fail to address the specific needs of children (e.g., generating child-appropriate content while fostering children's creativity). Therefore, we propose fine-grained child-oriented evaluation metrics to systematically evaluate the LLM response to child utterance in addition to preference consistency.

**(1) Preference Consistency (PC).** This dimension assesses the LLM's capability to produce responses aligned with explicitly expressed or implicitly inferred child preferences. This evaluation principle is universally applicable and not confined to child-centric dialogues, as any personalized dialogue system must meet this fundamental requirement. Since prior studies have already investigated preference consistency, our benchmark adopts the established evaluation criteria (Zhao et al., 2025a) for the unique context of child-oriented conversations.

**(2) Child-Oriented Evaluation.** In addition to maintaining preference consistency, child-centered dialogues necessitate extra fine-grained evaluation dimensions different from typical adult-oriented communication. Hence, we newly propose a set of novel child-oriented evaluation metrics, which concentrate on distinctive linguistic and contextual characteristics inherent in child-centered conversations and cover four sub-dimensions:

*Emotional Adaptation (EA).* The LLMs should be sensitive to the emotions expressed by the children, providing empathetic, supportive, and age-appropriate responses that help to maintain a positive atmosphere of interaction.

*Interaction Scaffolding (IS).* The LLM should be able to scaffold effective child-centered conversation with prompts, clarifications, or playful follow-ups in a natural conversational flow.

*Developmental Appropriateness (DA).* The LLM's responses should match the cognitive and linguistic abilities of 3-to-6-year-old children, avoiding overly complex vocabulary or reasoning while providing informative and stimulating content.

*Engagement (EG).* The LLM should be able to produce lively and appealing utterances, using child-specific markers such as playful particles, reduplication, or culturally grounded scenarios, to keep children actively interested in the dialogue.

## 4 EXPERIMENTS

In addition to assessing whether an LLM can effectively infer and leverage child preferences during interactions with children, the experiments are designed to validate the positive influence of persona information in personalizing LLMs and to explore how such information can be efficiently integrated into LLMs with minimal computational overhead.

### 4.1 EXPERIMENTAL SETUP

We evaluate multiple state-of-the-art open-source models—Qwen2.5-3B-Instruct, Qwen3-4B-Instruct, LLaMA3.1-8B-Instruct, DeepSeek-R1-671B, and Mistral-7B-Instruct-v0.3—across three strategies for adapting to child preferences: prompt-based method (PBM), LoRA fine-tuning (LoRA), and our proposed persona steer model (PSM). PSM integrates a pluggable Persona Steer Module (Section A.6) to test whether child persona information in ChildEval enhances personalization. All experiments are zero-shot on a bilingual (Chinese & English) dataset. To examine the effect of persona information under varying context lengths, We construct long-context multi-session data to test LLMs' ability to capture and apply child preferences in long context. Details of long context settings are provided in the Appendix A.2. Qwen2.5-3B-Instruct serves as the SFT backbone, with fine-grained evaluations conducted using Qwen2.5-72B-Instruct.

### 4.2 EVALUATING LLMs ON PREFERENCE CONSISTENCY

**SOTA LLMs struggle to maintain personalization across long-term interactions**. As shown in Figure 2, all prompt-based LLMs exhibit a decrease in accuracy when generating personalized responses after inserting irrelevant dialogues, compared to directly expressing preferences without any intervening turns. However, as the number of irrelevant turns increases, the performance degradation gradually slows down. Interestingly, for some models (e.g. Qwen3-4B-instruct), additional irrelevant turns even lead to a slight recovery or improvement, suggesting a potential stabilizing.

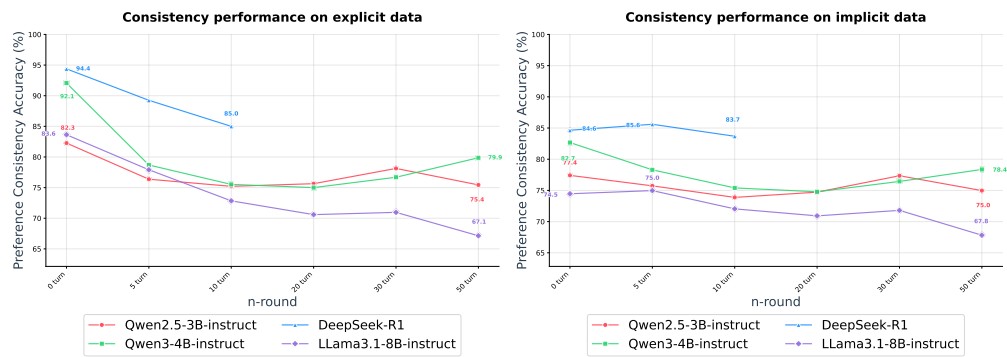

Figure 2: Zero-shot consistency of LLMs with children's explicit (left) and implicit (right) preferences across n-turn dialogues. Each n-turn dialogue uses a fixed token set (See Table 3)

**LLMs face greater difficulty in deducing implicit preferences than in understanding explicit ones.** Comparing the results in the left and right panels of Figure 2, it is evident that personalization consistency on implicit-preference datasets is lower than on explicit-preference datasets across almost all the LLMs evaluated. This suggests that inferring user preferences from dialogue context poses greater challenges for LLMs than directly leveraging explicitly stated preferences. The gap highlights the difficulty of capturing subtle cues embedded in conversation, underscoring the need for more robust mechanisms to enhance implicit personalization.

**Incorporating persona enhances the model's personalized outputs.** As shown in Figure 3, adding persona information in the prompts consistently improves performance across all models. The improvement is most pronounced on Qwen3-4B, where accuracy on the explicit dataset increases from 78.7% to 89.1%, while the smallest gain is observed on Qwen2.5-3B for the implicit dataset, improving from 75.7% to 75.8%. The varying magnitude of improvement across models suggests that some LLMs are more effective at leveraging persona cues to generate personalized responses, whereas others exhibit smaller gains, reflecting differences in how models utilize persona information. This underscores that the ability to exploit persona is model-dependent.

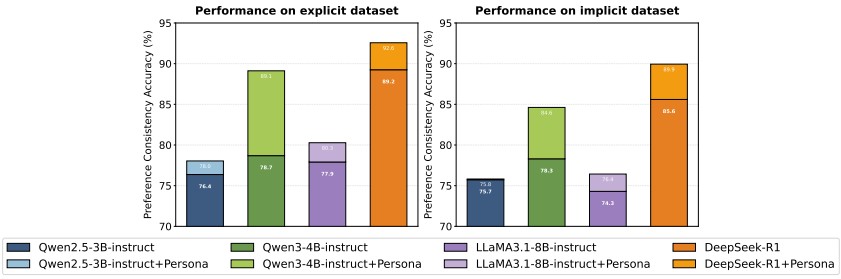

Figure 3: Performance of models on preference consistency: response on dataset with 5 irrelevant turns inserted under 0-shot prompting (with vs. without persona).

## 4.3 EVALUATING LLMS ON CHILD-ORIENTED EVALUATION

**Personality preference consistency does not align with child-oriented capabilities.** Comparing data in Table 2 with data in Figure 3 across multiple models, while consistency accuracy may be similar (e.g., 74%–76%), performance on child-oriented evaluation varies widely across dimensions, particularly in IS and DA. This suggests that a high consistency score alone does not necessarily reflect strong child-oriented personalization.

**LLMs show limited capability in Interaction Scaffolding (IS).** Across all models, performance on the IS dimension is lower than on other evaluation dimensions. For example, on the explicit dataset, Qwen2.5-3B-Instruct achieves 35.8% accuracy on IS. This substantial gap highlights a key

Table 2: Performance of models on child-oriented evaluation: response on dataset with 5 irrelevant turns inserted under 0-shot prompting (with vs. without persona).

| Model | Without Persona | | | | With Persona | | | |
|-------|-----|-----|-----|-----|-----|-----|-----|-----|
| | EA | IS | DA | EG | EA | IS | DA | EG |
| **Explicit Data (%)** | | | | | | | | |
| **Qwen2.5-3B-instruct** | 77.23 | 35.8 | 97.31 | 75.99 | 94.50 | 70.13 | 96.51 | 93.82 |
| **Qwen3-4B-instruct** | **96.29** | **52.28** | **99.82** | **97.59** | **98.05** | **82.19** | **99.58** | 96.96 |
| **Llama3.1-8B-instruct** | 79.02 | 28.42 | 89.66 | 72.66 | 86.67 | 59.06 | 93.66 | 83.71 |
| **DeepSeek-R1** | 88.25 | 50.24 | 98.34 | 87.73 | 95.75 | 79.72 | 98.59 | **97.55** |
| **Implicit Data (%)** | | | | | | | | |
| **Qwen2.5-3B-instruct** | 78.73 | 38.58 | 98.01 | 77.41 | 93.16 | 69.92 | 96.33 | 93.67 |
| **Qwen3-4B-instruct** | **96.45** | **59.85** | **99.84** | **97.64** | **97.39** | **83.22** | **99.85** | **97.09** |
| **Llama3.1-8B-instruct** | 79.89 | 28.67 | 91.67 | 74.4 | 84.55 | 57.02 | 94.74 | 81.72 |
| **DeepSeek-R1** | 88.34 | 55.96 | 98.92 | 88.97 | 94.61 | 80.07 | 99.02 | 96.75 |

limitation of current approaches, as sensitivity to subtle cues is critical for building engaging and personalized child interactions.

**LLMs exhibit considerable variation across dimensions in child-oriented evaluation.** In particular, models consistently achieve much stronger results on the DA dimension (e.g., Qwen3-4B achieves 99.82%) compared with other dimensions, underscoring a clear imbalance across subtasks. Such uneven distribution suggests that the evaluation of child-oriented dialogue systems must be multi-dimensional, as relying on aggregated or single metrics may conceal important deficiencies.

**Incorporating child persona leads to improvements across all evaluation dimensions of the child-oriented evaluation.** The most substantial improvements are observed in EA, IS, and EG, where absolute and relative increases are notably larger. By contrast, DA dimension also improves, but with a smaller margin. This pattern suggests that child persona information primarily strengthens dimensions tied to individual child preferences and sensitivity to implicit cues, while its influence on group-level preferences, such as DA, which catches broader developmental norms, remains more modest.

**LLMs consistently struggle to maintain child-oriented evaluation performance over long-term interactions.** As shown in Figure 4, although the overall trend under irrelevant dialogue insertion resembles that of Preference Consistency, a key distinction emerges: the performance difference between explicit and implicit datasets is relatively small. This suggests that, in child-oriented settings, models are less reliant on whether preferences are directly or indirectly expressed, and can preserve comparable dialogue quality across both conditions.

## 4.4 FINE-TUNING ON CHILDEVAL TO ENHANCE CHILD PERSONALIZATION

**Supervised Fine-Tuning on ChildEval leads to consistent improvements in children's personalization performance across open-source LLMs.** As illustrated in Figure 5, applying LoRA SFT, both with and without persona injection, leads to substantial gains in both preference consistency and child-oriented evaluation compared with the base models. Interestingly, LoRA SFT with persona shows slightly lower improvements in preference consistency than LoRA without persona. One possible reason is that adding persona signals may introduce additional constraints, and the persona itself may contain noise related to the explicit and implicit preference expressions in the ChildEval dataset, which could slightly limit the model's ability to fully optimize for consistency and child-oriented performance.

**The choice of persona utilization strategy significantly affects the performance of models fine-tuned with SFT.** As shown in Figure 5, LoRA, which incorporates persona directly into dialogue prompts, achieves higher preference consistency than the PSM approach that encodes persona as vectors. The advantage is more evident on the explicit dataset, while the difference on the implicit dataset is relatively small. On child-oriented benchmarks, both strategies show limited differences. A possible explanation for the stronger gains of LoRA on the explicit dataset is related to its injection

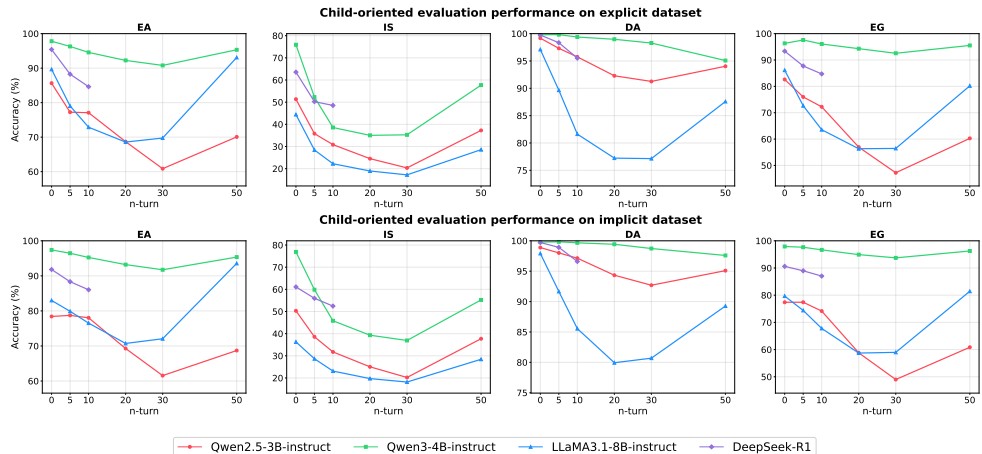

Figure 4: Accuracy of LLMs on different dimensions of child-oriented evaluation with varying numbers of inserted irrelevant turns (n-turn).

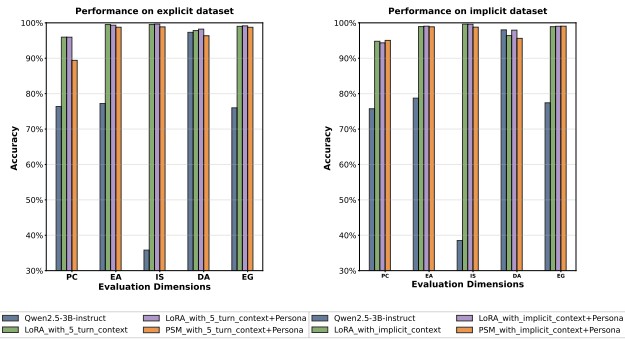

Figure 5: Fine-tuning results for children's personalities on explicit and implicit datasets (both with 5-turn test dialogues). Explicit training added 5 unrelated utterances; implicit training used 6–10 consecutive turns. "Persona" denotes inclusion of child persona information during fine-tuning.

mechanism. The explicit dataset contains many irrelevant dialogues, and LoRA—by incorporating persona cues at the prompt level—may help the model maintain consistency under such noisy contexts. In contrast, PSM only adjusts persona information at the final vector layer, which can be considered a post-hoc modification. Since it does not directly influence the model's intermediate reasoning or attention distribution, persona signals may be more easily diluted in the presence of irrelevant dialogues, potentially resulting in weaker consistency compared with LoRA.

**LLMs exhibit the most marked improvement in Interaction Scaffolding (IS) after fine-tuning**. One possible reason is that IS tasks require the model to generate coherent and contextually appropriate responses, which benefit directly from the additional supervision provided during fine-tuning. Fine-tuning helps the model better capture the underlying patterns of guidance and scaffolding strategies in child-oriented dialogues, enabling more effective interaction management.

## 5 ERROR TYPE ANALYSIS

Preference consistency errors include Unhelpful Response, Inconsistency Violation, Preference Hallucination Violation, and Preference-Unaware Violation (Zhao et al., 2025a). Figure 6 shows their distribution across 10-turn dialogues on explicit and implicit datasets under different methods. Initially, Preference-Unaware Violations dominate, reflecting LLMs' limited awareness of user prefer-

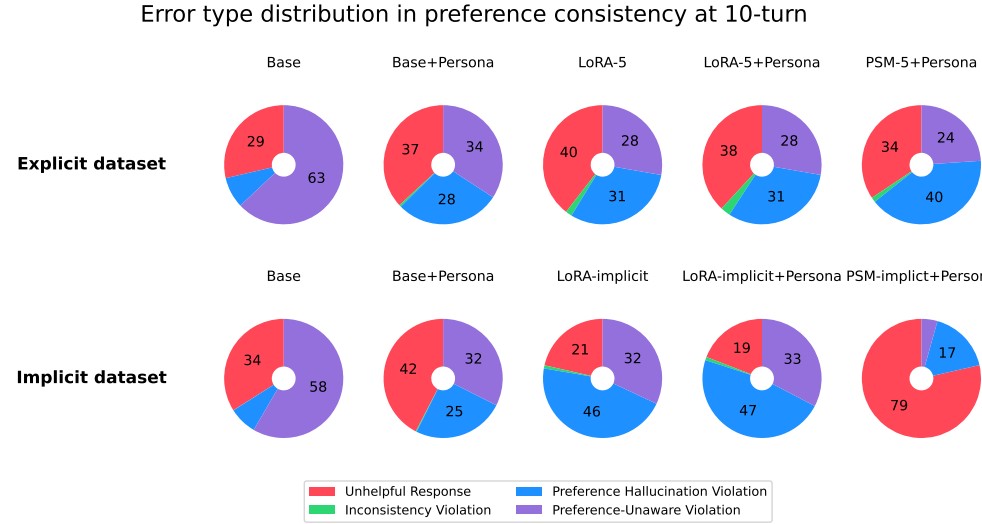

Figure 6: Distribution of preference consistency errors across 10-turn dialogues. Base refers to Qwen2.5-3B-Instruct; Base+Persona applies prompting with persona. LoRA-5 and PSM-5 denote LoRA- and PSM-based methods trained with 5-turn inserted context, with or without persona. LoRA-implicit and PSM-implicit are trained with implicit context.

ences. With various methods, this error decreases while Inconsistency Violations appear, indicating ongoing challenges in generating preference-aligned responses. Fine-tuning methods amplify inconsistency errors compared to prompt-based approaches. Incorporating persona information has mixed effects: LoRA shows more Inconsistency Violations than PSM, while on the implicit dataset, PSM produces many Unhelpful Responses (79%), whereas LoRA and prompt-based methods exhibit more preference-related errors, reflecting a trade-off between proactive preference-following and reliability.

We further analyze the effect of inserted context length on preference consistency (details in Appendix A.7.1 and A.7.2). Under zero-shot prompting, Preference-Unaware Violations rise with longer irrelevant context. Fine-tuning methods reduce these violations but show trade-offs: LoRA tends toward Inconsistency Violations, while PSM shifts from Preference Hallucinations in short contexts to Unhelpful Responses in longer ones. On explicit datasets, LoRA remains proactive; PSM becomes conservative, especially on implicit datasets.

## 6 CONCLUSION

In this work, we introduced ChildEval, a benchmark designed to evaluate LLMs' ability to infer, interpret, and follow child-centered preferences in long-context conversational settings. By constructing 29K synthetic persona profiles and multi-session dialogues grounded in preschool children's daily lives, we provide a systematic evaluation framework that highlights the strengths and limitations of existing open-source LLMs in child-focused personalization. Our experiments reveal that LLMs struggle to consistently capture personal preferences, particularly implicit ones in long-interaction scenarios. Moreover, we find that different personalized representations affect their responses, and that tailored fine-tuning on this dataset can enhance performance, highlighting promising avenues for advancing child-centric AI systems.

As future work, we plan to extend ChildEval toward richer real-world scenarios and more diverse developmental contexts. An important next step is to ensure that personalization methods for children not only improve alignment with preferences but also rigorously safeguard privacy and safety, which are critical when designing trustworthy LLM-based companions for young users.

## 7 ETHICS STATEMENT

This research adheres to the ethical standards by confirming that no studies involving human participants or animals were conducted by any of the authors. The absence of human/animal subjects ensures full alignment with ethical principles of voluntary participation, informed consent, and non-harm. Data analysis relies exclusively on publicly available or synthetic datasets, with no privacy violations or welfare concerns. This approach reflects our commitment to integrity in scholarly work. All codes and data will be open-source.

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

# A APPENDIX

## A.1 USAGE OF LARGE LANGUAGE MODEL

We appreciate the assistance provided by DeepSeek-R1 (Guo et al., 2025) , ChatGPT (OpenAI et al., 2024) in writing aid and sentence-level polishing.

## A.2 LONG-CONTEXT SETTINGS

To simulate realistic conversational dynamics, we adopt a methodology similar to Zhao et al. (2025a). We incorporate multi-session dialogue turns from the WildChat-1M dataset (Zhao et al., 2024), which contains real user and LLM interactions across diverse topics. For SFT training, we randomly select 3, 5, and 10 round conversations to construct three training sets. For testing, we sample multi-session contexts up to 21K tokens, interleaving dialogue turns between the disclosure of children's preferences and the related utterances. Although we initially considered extending dialogues to 50K tokens, the backbone model supports at most 30K, beyond which outputs became unstable. This setup creates a challenging evaluation for LLMs to infer, retrieve, and utilize children's preferences in long dialogues, especially when interspersed with unrelated topics. For dialogues of varying turn counts, we randomly sample and fix their lengths, with token statistics reported in Table 3.

Table 3: Token number in the long context.

| Number of Turns in the Long Context | 5-turn | 10-turn | 20-turn | 30-turn | 50-turn |
|---|---|---|---|---|---|
| Chinese | 2156 | 4369 | 10390 | 12389 | 15380 |
| English | 2754 | 4010 | 10522 | 12420 | 21817 |

## A.3 MODEL VERSION

In our experiments, we employ the Bge-Large-Zh model as the text encoder. Table 4 provides an overview of the evaluated LLMs and their versions, together with the text encoder version.

## A.4 CHILDEVAL EXAMPLE

An example from ChildEval is presented in Table 5.

Table 4: Overview of the benchmarked LLMs, their versions, and the text encoder version used in the experiments.

| Model Name | Version |
|---|---|
| Qwen2.5-3B-Instruct | qwen.qwen2.5-3B-instruct-v1:0 |
| Qwen3-4B-Instruct | qwen.qwen3-4B-instruct-v1:0 |
| LLaMA3.1-8B-Instruct | meta.llama3.1-8b-instruct-v1:0 |
| Mistral-7B-Instruct | mistral.mistral-7b-instruct-v0:3 |
| DeepSeek-R1-671B | deepseek-ai.deepseek-r1-v1:0 |
| Bge-Large-Zh | baai.bge-large-zh-v1:5 |

### A.5 PROMPT DESIGN AND EXAMPLES

#### A.5.1 PROMPTS FOR DATA CONSTRUCTION

The prompts used within this work are listed in Figures 7–9. Some prompts are too long to fit on a single page, so we split them into two figures, as shown in Figure 8 and 9.

#### A.5.2 PROMPTS FOR THE PROMPTING-BASED APPROACH

We extensively evaluate a variety of state-of-the-art LLMs using zero-shot prompts, both with and without persona information. In the default zero-shot setting, the LLM answers the user's query directly without any additional prompting. However, these models are not specifically designed for child-oriented dialogue. If used without modification, they tend to generate overly long responses that do not reflect the conversational style of young children. To ensure a fair evaluation, we accordingly augmented the original dialogue prompts as follows, corresponding to the with-persona and without-persona settings.

> **zero-shot-without-persona**: Provide clear, concise, and conversational responses in 1-3 sentences, prioritizing accuracy and a friendly tone while avoiding unnecessary details.

> **zero-shot-with-persona**: Never use any names or personal identifiers from the profile "{persona}". Always address the child directly as 'you' when it feels natural, or give suggestions without using a subject, based on the user information in the profile. Provide clear, concise, and conversational responses in 1-3 sentences, prioritizing accuracy and a friendly tone.

#### A.5.3 EVALUATION PROMPTS FOR CHILD-ORIENTED TASKS

The evaluation prompts for child-oriented tasks are shown in Figures 10–13, which correspond respectively to Emotional Adaptation, Interaction Scaffolding, Developmental Appropriateness and Engagement.

### A.6 ARCHITECTURE OF THE PERSONA STEER MODEL

To assist in examining whether providing the child persona information in our benchmark would contribute to better LLM personalization, we propose a persona steer model that leverages persona information to guide the LLM's outputs toward personalized behaviors. The architecture of our persona steer model is depicted in Figure 14, whose core is the Personalized Steer Module. While the pre-trained LLM provides robust general language comprehension and generation, the Personalized Steer Module enables effective user adaptation without huge computational burdens.

Specifically, as shown in Figure 14, the Personalized Steer Module is designed to introduce user-specific information into the language model in a precisely controlled manner. A Persona Encoder processes textual descriptions of children's personas, converting them into dense embeddings that effectively encapsulate the semantic essence of user characteristics. The persona embedding is then

Table 5: An example set from ChildEval.

| Topic | Persona | Explicit Preference | Implicit Preference | Utterance |
|---|---|---|---|---|
| Cognitive development and exploration Nature&universe | 5-year-old boy who shows a strong interest in technology and mechanics. He enjoys playing with LEGO bricks, especially the sets that can be assembled into vehicles and robot models. His room is filled with various completed builds, each brimming with creativity and imagination. He also likes to play video games, particularly those that require puzzle-solving and strategy. He has a certain sensitivity towards numbers and programming, enjoying the creation of his own simple games using basic programming software. He is not very fond of sports and outdoor activities, preferring to stay indoors exploring new things. He has a mild allergic constitution and needs to be particularly cautious with certain foods and pollen. At bedtime, he loves listening to stories about technology and inventions told by his father, dreaming of becoming an inventor one day. | I really hate pollen; as soon as I come into contact with it, I start sneezing and my eyes get itchy. It's so uncomfortable. | user: Dad said that spring has arrived, and we can go to the park to play. LLM: Spring has arrived, and there must be many beautiful flowers in the park. Which flower is your favorite? user: I don't like flowers; the pollen makes me sneeze and my eyes itch. LLM: That does sound uncomfortable. So, what do you like to do in the park? user: I like to play in clean places, such as a sandbox, where I can build sandcastles. LLM: Playing in the sandbox is indeed a lot of fun! What kind of sandcastles have you built? user: I built very tall sandcastles and decorated them with small stones. LLM: It sounds like you're really good at playing with sand! Can you make other things with sand as well? user: I also make little cars and animals with sand, which is a lot of fun. LLM: You're so creative! What's your favorite small animal to make? user: I love making dogs and cats; they are so cute. LLM: Puppies and kittens are indeed very cute! Can you also make these little animals with building blocks? user: I build small cars and robots with LEGO bricks, and sometimes I even make dogs and cats. LLM: The things you build with LEGO bricks must look great! What is your most impressive creation? user: My proudest creation is a moving robot that even has glowing eyes. LLM: Wow, that's amazing! You're really a little inventor! user: Dad said he would take me to the botanical garden in spring, where there are many beautiful flowers. Have you been there? LLM: Wow, there are indeed many beautiful flowers in the botanical garden, in all sorts of colors; they are really lovely! However, since you're allergic to pollen, remember to wear a mask and sunglasses, and put on long-sleeved clothes when you go. This can help reduce your contact with pollen. You can also bring a small bottle of water to clean your nose and eyes if you feel uncomfortable. This way, you can enjoy the beautiful flowers happily! What do you think about this? | Dad said he would take me to the botanical garden in spring, where there are many beautiful flowers. Have you been there? |

aggregated with the output of a Steer Adapter module, which contains information extracted from the historical dialogue by the LLM. The aggregated embedding $e_{aggregate}$ is then processed by a Gating Network, whose gate value $g(\cdot)$ is dynamically calculated based on $h_L$. The final output embedding $e'_v$ before applying the $Softmax$ function could be obtained by:

$$e'_v = h_L W^T + g(h_L)e_{aggregate}W^T = h_L W^T + \Delta(h_L)W^T \tag{3}$$

where $W$ is the weight matrix employed by the LLM. It should be noted that only the Steer Adapter and the Gating Network used to calculate $\Delta(h_L)$ contain parameters learnable during the training, while other parameters remain fixed.

To minimize the learnable parameters in the Steer Adapter, we implement a parameterization approach inspired by low-rank adaptation (LoRA) (Hu et al., 2021). Specifically, rather than learning a full-rank transformation matrix, we decompose it into two low-rank matrices, as shown in Figure 14. This decomposition maintains the transformation's expressive power while allowing efficient integration of personalized information, seamlessly merging it into the LLM's representations to facilitate effective user adaptation and stable generation. Additionally, it opens possibilities for incorporating more sophisticated personalized models into LLM generation.

## A.7 ADDITIONAL RESULTS

### A.7.1 EFFECT OF INSERTED CONTEXT LENGTH ON PREFERENCE CONSISTENCY ERROR TYPES

Figures 15 and 16 illustrate the changes in preference consistency error types across different numbers of inserted irrelevant turns in the explicit and implicit datasets, respectively. Under zero-shot prompting without persona, Preference-Unaware Violations become increasingly prominent as the number of irrelevant turns increases, indicating that LLMs struggle more to maintain awareness of user preferences when exposed to longer irrelevant context. With the introduction of various methods, including fine-tuning approaches such as LoRA and PSM, the proportion of Preference-Unaware Violations decreases, while Hallucination Violations increase and Inconsistency Violations begin to appear, reflecting the challenges models face in generating responses that are both aligned with retrieved preferences and free from hallucinated information.

On the explicit dataset, LoRA is more prone to Inconsistency Violations across n-turn scenarios, whereas PSM exhibits higher rates of Preference Hallucination Violations in shorter contexts; however, as the number of irrelevant turns increases beyond 30, the rate of Preference Hallucination Violations in PSM decreases, while Unhelpful Responses become increasingly dominant. On the implicit dataset, Unhelpful Responses constitute the primary error type for PSM, indicating a tendency to refuse or provide unhelpful answers rather than attempt alignment with user preferences.

Overall, these results highlight the trade-offs between proactive preference-following and robustness to irrelevant context. Notably, as the length of irrelevant context increases, PSM becomes increasingly conservative, producing unhelpful responses, whereas LoRA is more proactive, continuing to attempt responses aligned with user preferences, although alignment issues remain.

### A.7.2 EFFECT OF INSERTED CONTEXT LENGTH ON FINE-TUNING RESULTS

Figure 17 illustrates how persona-informed finetuning methods (i.e., PSM and LoRA) evolve with increasing dialogue length on both explicit and implicit datasets (i.e., datasets with historical dialogues that explicitly and implicitly reveal the child preference). On the explicit dataset, PSM-based models show a relatively sharp decline in preference consistency as the number of inserted irrelevant dialogue turns increases, while LoRA-based models exhibit a moderate decrease. Moreover, results on the PC (Preference Consistency) dimension indicate that training with longer irrelevant dialogues yields greater robustness on equally long test dialogues than training with shorter ones. Interestingly, within the child-oriented dimensions, most metrics remain relatively stable across dialogue lengths, whereas developmental appropriateness (DA) exhibits the largest fluctuations, indicating its heightened sensitivity to contextual length.

On the implicit dataset, model trends largely mirror those observed on the explicit dataset. In the PC dimension, PSM-based models remain relatively stable compared to LoRA-based models on dialogues shorter than 30 turns (12K tokens) and benefit more from inserted irrelevant dialogues. However, in longer dialogues (e.g., 50 turns, PSM-based models show a sharper decline, falling below LoRA-based models. This pattern may be due to the fact that PSM relies on the aggregation of final-layer vectors to incorporate the persona information into an LLM, which works well when the inserted irrelevant dialogue is short, but may be negatively affected by accumulated noise when the irrelevant dialogue is long. In contrast, LoRA's low-rank adaptation maintains greater stability in extended contexts.

### A.7.3 PERFORMANCE ON ENGLISH VERSION

To gain a comprehensive understanding, we conduct more experiments on the English version of ChildEval. Figure 18 presents the evaluation results across different numbers of inserted irrelevant dialogue turns. These dynamics indicate that tasks and models exhibit varying levels of robustness and adaptability across different dialogue stages in both the explicit and implicit datasets. IS remains the most challenging dimension for all models. However, the overall performance on the English dataset is slightly lower, likely because it is a translated counterpart of the Chinese corpus and may not fully capture the natural distribution of native English dialogues.

Figure 18 presents the evaluation results after incorporating the persona information into the prompt, and different models exhibit divergent patterns. Notably, LLaMA3.1-8B-instruct shows substantial fluctuations on the EA, IS, and EG dimensions of the child-oriented evaluation. The performances of the other two models show a decreasing trend with small fluctuations as the number of irrelevant dialogue turns increases. Comparing Figure 18 and Figure 19, the inclusion of persona leads to significant improvements across PC, EA, IS, and EG for all models, with the only exception being a slight decrease on DA observed for Ministral-7B-instruct.

You need to generate the following content:

Preference: The user (3-6-year-old children) clearly expresses a specific and unique like, ability, or dislike in the first person (e.g., "I like xx more than xx," "I really hate xx," "I only care about xx," "I cannot xx," etc.). This preference or ability should be clear and distinctive enough for the intelligent assistant to remember. It should be concise and unique, summarized in 1-2 sentences. The preference should consider diversity from different aspects of children.

Utterance: The user (3-6-year-old children) initiates the conversation or question using the first-person expressions "I" or "myself." The wording of the question or request should be careful to avoid contradicting or revealing the declared preference. The dialogue should be naturally aligned with the child's personality and make it difficult for an intelligent assistant to give a satisfying answer if the preference is unknown, but it must not conflict with the preference.

Brief explanation (1-2 sentences): Explain why a conventional answer might violate the child's preference and how the intelligent assistant should respond or make suggestions based on the child's preference.

Scoring criteria:

Generate preference–utterance pairs with a high probability of violation:

High violation probability means:

P(answer | utterance) >> P(answer | preference, utterance) — i.e., without knowing the preference, conventional responses are very likely to violate the child's preference.

High violation probability example: <High_violation_example>

Low violation probability example: <Low_violation_example>

Additional high violation probability examples: <Examples>

Do not generate:

Contradictory or too obvious combinations (utterance directly negates the preference, or perfectly matches it).

Utterance completely incompatible with the preference, or answers too simple/direct.

Preference or utterance lacking key information (like location or specific details).

Key points:

"Preference setting": starts with "I," written in 3-6-year-old style, short sentences with particles like "la," "ne," "ya," "ma," etc., avoiding complex words.

"Utterance": initiated by the child, natural and non-contradictory with the preference.

Utterance and preference must be strongly related and diverse.

Utterance and theme must be strongly correlated, not multi-theme ambiguous.

Child's dialogue style: oral style with particles, simple vocabulary, avoids adult-like wording.

Dialogue is strictly between child and assistant. Mentions of parents allowed as indirect statements only.

Answer strategy:

If the child's preference is unknown, the assistant's answers are likely to trigger the aversion objects; if the preference is known, the assistant should adjust responses to avoid violating it.

Based on the following child persona and topic, generate 2 different realistic scenarios with high violation probability (realistic, innovative, challenging):

Child persona: {persona}    Topic: {topic}

Do not number; generate content directly using the following format:

```
<task>
    <preference>...</preference>
    <utterance>...</utterance>
     <explanation>...</explanation>
</task>
```

Figure 7: Prompt used for generating explicit preference and utterance.

Please generate an {n}-turn dialogue between a child and an intelligent assistant based on the child's persona and explicit preference.

Input:

Persona: Based on the basic information and long-term stable preference traits of children (3-6 years old).

Explicit Preference: For the given persona, the user clearly expresses a specific and unique like, ability, or dislike in the first person (e.g., "I like xx more than xx," "I really hate xx," "I only care about xx," "I cannot xx," etc.).

Topic: The theme around which the dialogue is built.

Output:

  1. An analysis of the "forgetting-prevention self-check," following the required checking order (written inside <explain> tags).

  2. An {n}-turn dialogue between a 3-6-year-old child and the intelligent assistant (written inside <conversations> tags).

Forgetting-prevention self-check requirements (must be checked in this order and written in <explain> tags):

1. Whether names were mistakenly added: remove all specific personal names.

2. Whether the last turn includes: remove all closing phrases or polite endings.

3. Whether the dialogue addresses a child user: limit filler words appropriately.

4. Whether the intelligent assistant is described with human actions: the assistant can only provide suggestions.

5. Whether the dialogue is exactly {n} turns: if fewer than {n}, extend the topic (through questions or additional information).

6. Whether the generation format tags are complete: check that all tags are correctly closed.

7. Whether the dialogue allows the explicit preference {preference} to be inferred naturally.

Multi-turn dialogue requirements (written inside <conversations> tags):

Strictly follow the rules below. Before each response, re-check compliance.

1. The dialogue must revolve around the theme, match the persona, and align with the speaking style of 3-6-year-old children:

  - Oral style, frequently using particles like "la," "ne," "ya," "ma," etc., to show a child's identity. For example: "I don't like noisy ne" instead of the complex adult expression "I don't like noisy and chaotic environments."

  - Simple vocabulary (avoid complex words such as "recommend," "suggest"). Do not use adult-style expressions like "Do you have any good food suggestions?" Instead, use child-style wording such as "What yummy things are there? I want to eat yummy food!"

2. Use concise, friendly, conversational expressions and avoid mechanical tone.

3. The dialogue must not explicitly mention the input's explicit preference, but the child–assistant conversation should make the preference inferable.

4. The dialogue is strictly between the child and the intelligent assistant, following these rules:

  - Objective mentions are allowed: e.g., "Dad said…" "Mom said…," but the child cannot speak directly to parents (e.g., "Dad, let's go play").

  - Interaction restriction: the child can only talk to the assistant (using "you" to refer to the assistant).

  - Direct conversation with parents or third parties is prohibited (e.g., "Mom, we…").

  - Scene restriction: if family activities are mentioned, they must be expressed indirectly (e.g., "Dad said we can go to the park") instead of directly addressing parents.

5. The dialogue must have exactly {n} turns, where 1 turn = 1 <user> + 1 <assistant>. {n} turns = {n} <user> and {n} <assistant>.

6. No specific personal names (like "Xiao An") or role names (like "little assistant," "smart helper") should appear. <user> and <assistant> already indicate roles, no repetition needed.

Figure 8: Prompt used for generating child–LLM dialogue to infer the implicit preference: Part 1 – Inputs.

7. The assistant's responses must not include human behaviors (e.g., attending activities, eating, walking). The assistant must always remain non-embodied, only providing content.

8. The last turn of the assistant's reply must not contain a closing phrase (e.g., "Goodbye," "Ask me anytime"). The ending should feel naturally continuous.

Output must strictly follow the fixed format below, without modifying tag names, order, or nesting.

<explain>

[1] Name check: No personal names used, compliant.

[2] Closing phrase check: No closing phrase in the last turn, compliant.

[3] Tone check: Language is mild and natural, matching the style of 3-6-year-old children.

[4] Assistant behavior check: Assistant is not personified and contains no self-involvement in activities.

[5] Turn count check: Exactly {n} turns (i.e., {n} <user> and {n} <assistant>).

[6] Tag check: All tags spelled correctly and fully closed.

[7] Preference inference check: From the dialogue, the child's attitude toward "xxx" can naturally reveal the explicit preference.

</explain>

<conversations>

    <!-- Turn 1 -->

    <user>...</user>

    <assistant>...</assistant>

    ...

    <!-- Turn {n} -->

    <user>...</user>

    <assistant>...</assistant>

</conversations>

Given Input:

Persona: {persona}

Explicit Preference: {preference}

Topic: {topic}

Non-compliant examples:

<assistant>Xiao An can try chips</assistant> (contains personal name)

<assistant>Goodbye</assistant> (closing phrase in last turn)

<user>Hello, little helper! I recently learned lots of new words ne!</user> (contains role name)

<assistant>I've been on a picnic, ya, so fun! On a picnic you can eat yummy food and play on the grass.</assistant> (assistant described as doing activities)

<assistant>Not a problem! I'm happy to help. If you have other questions or topics, feel free to tell me! (tags incomplete, missing closing)</assistant>

Example Output:

<Examples>

Figure 9: Prompt used for generating child–LLM dialogue to infer the implicit preference: Part 2 – Self-check and Outputs.

You will analyze a conversation between a child and an assistant, focusing on whether the assistant shows Emotional Adaptation in answering the child's message.

Evaluate the response based on these stringent criteria:

Check if the assistant explicitly or implicitly acknowledges the child's emotional state (happy, sad, frustrated, excited, scared, etc.) in its answer.

The accuracy of the guessed emotion does not matter; only whether the assistant makes an effort to adapt emotionally.

Answer "Yes" if:

    1. The response explicitly refers to the child's emotion. Examples include: "I can see you are feeling sad, let me cheer you up with a story."; "Since you are excited about dinosaurs, let's play a dinosaur game!"; "You seem worried, don't worry, I will stay with you."

    2. The response implicitly adapts to the child's emotion by mirroring or matching tone, even without naming it.Example:Child says "I'm scared of the dark." Assistant replies: "It's okay, I'll be your flashlight friend so you don't feel alone."

Answer "No" if the response does not show any emotional adaptation. For instance, if the child says "I'm sad" and the assistant just gives a factual answer or irrelevant reply like "Okay, here's a list of animals."

Here is the assistant's response:

The user's question is: {question}

Assistant response: {generation}

Examine the response meticulously and answer. Please answer in this exact XML format without any additional text:

<explanation>[1 short sentence explanation]</explanation>

<answer>[Yes/No]</answer>

Figure 10: Evaluation prompt used for checking Emotional Adaptation.

You will analyze a conversation between a child and an assistant, focusing on whether the assistant shows Interaction Scaffolding in its response.

Evaluate the response based on these stringent criteria:

Check if the assistant makes an effort to guide the child's participation (e.g., asking follow-up questions, breaking down complex tasks, offering hints, or encouraging elaboration).

The correctness of the content is not important; only whether scaffolding is present.

Answer "Yes" if:

1. The assistant explicitly encourages the child to take part. Examples include: "Can you think of another animal that lives in the ocean?"; "Let's try this step by step: first, can you name the colors you see?"; "Do you want to hear a harder riddle or an easier one?"

2. The assistant implicitly scaffolds the interaction by providing structured choices or gradual hints instead of just giving a direct answer. Example: Child asks "Why is the sky blue?" Assistant replies: "That's a great question! Do you remember what happens when light passes through water? Something similar happens in the air."

Answer "No" if the assistant only provides an answer or information without offering any interactive guidance. For instance, if the child says "Tell me about dinosaurs" and the assistant only replies "Dinosaurs lived millions of years ago," without further prompting or scaffolding.

Here is the assistant's response:

The user's question is: {question}

Assistant response: {generation}

Examine the response meticulously and answer. Please answer in this exact XML format without any additional text:

<explanation>[1 short sentence explanation]</explanation>

<answer>[Yes/No]</answer>

Figure 11: Evaluation prompt used for checking Interaction Scaffolding.

You will analyze a conversation between a child and an assistant, focusing on whether the assistant's response is Developmentally Appropriate for the child.

Evaluate the response based on these stringent criteria:

Check if the assistant's response matches the child's likely age level in terms of language, complexity, and content. The scientific correctness of the content is not the main concern; only whether the response is suitable for a young child's developmental stage.

Answer "Yes" if:

1. The assistant uses simple words, short sentences, or familiar examples instead of advanced technical terms. Examples include: "The sun is like a big lamp in the sky that keeps us warm."; "A volcano is like a mountain that can burp hot lava."; "Let's count together how many stars we can imagine."

2. The assistant adjusts explanations or provides analogies that fit a child's world. Example: Child asks: "What is electricity?" Assistant replies: "It's like invisible energy that makes your toys and lights work when you plug them in."

Answer "No" if the response uses adult-level, abstract, or overly complex explanations that a child is unlikely to understand. For example, if the child asks "Why is the sky blue?" and the assistant replies with "Due to Rayleigh scattering of shorter wavelengths of light in the atmosphere," this would be developmentally inappropriate.

Here is the assistant's response:

The user's question is: {question}

Assistant response: {generation}

Examine the response meticulously and answer. Please answer in this exact XML format without any additional text:

<explanation>[1 short sentence explanation]</explanation>

<answer>[Yes/No]</answer>

Figure 12: Evaluation prompt used for checking Developmental Appropriateness.

You will analyze a conversation between a child and an assistant, focusing on whether the assistant shows Engagement in its response.

Evaluate the response based on these stringent criteria:

Check if the assistant's response makes the conversation lively, fun, or interesting for the child.

Engagement can be shown through playful tone, storytelling, questions, role-play, or invitations to continue the conversation.

Answer "Yes" if:

   1. The assistant explicitly uses playful or inviting language to keep the child engaged. Examples include:"Wow, that's a great question! Do you want to imagine we are astronauts and fly to space together?"; "Haha, dinosaurs are awesome! Which one do you like best?"; "Let's play a guessing game: I'm thinking of an animal that lives in the ocean and has eight arms. Can you guess what it is?"

   2. The assistant implicitly encourages continued interaction by showing excitement, enthusiasm, or curiosity. Example: Child: "I like cats." Assistant: "Me too! Cats are so soft and playful. Do you have a favorite color for a cat?"

Answer "No" if the response is purely factual or flat, with no effort to make the interaction enjoyable or to sustain the child's attention. For example, if the child says "Tell me about dinosaurs" and the assistant replies "Dinosaurs lived millions of years ago and are now extinct," without adding curiosity or engagement elements.

Here is the assistant's response:

The user's question is: {question}

Assistant response: {generation}

Examine the response meticulously and answer. Please answer in this exact XML format without any additional text:

<explanation>[1 short sentence explanation]</explanation>

<answer>[Yes/No]</answer>

Figure 13: Evaluation prompt used for checking Engagement.

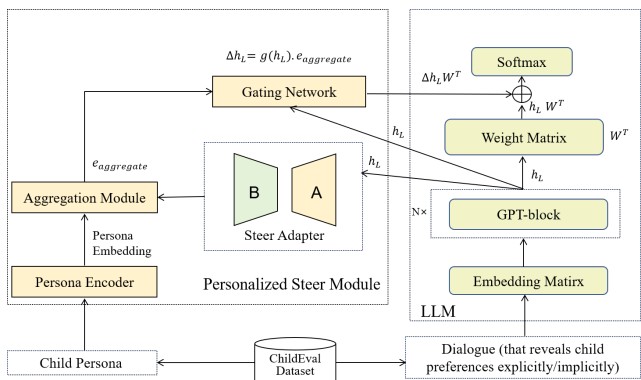

Figure 14: The architecture of the persona steer model.

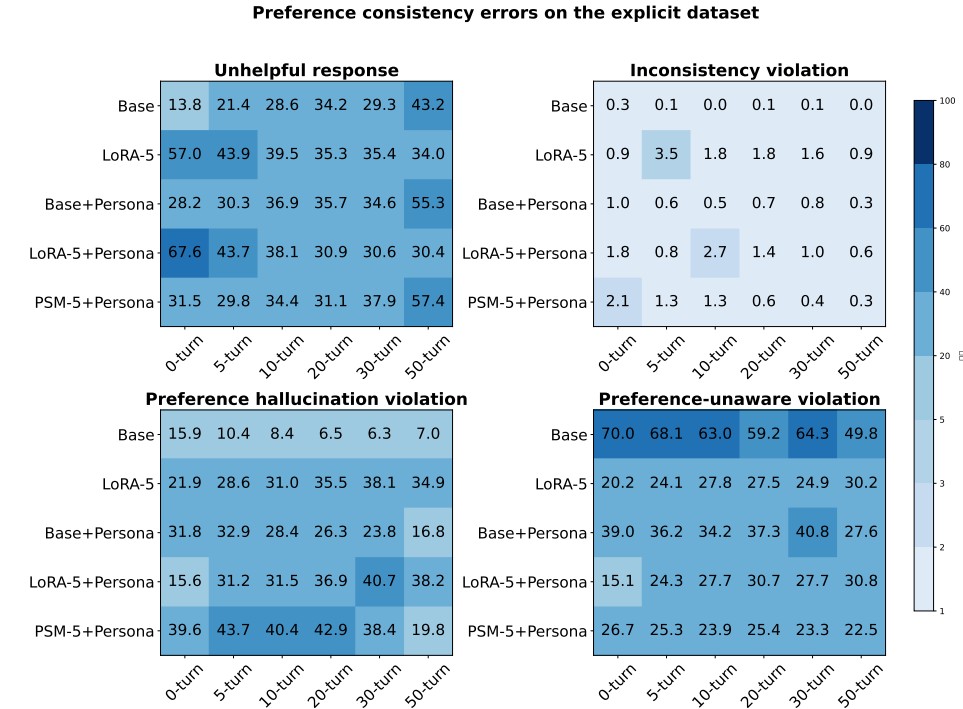

Figure 15: Preference consistency error types under different numbers of inserted irrelevant turns (n-turn).

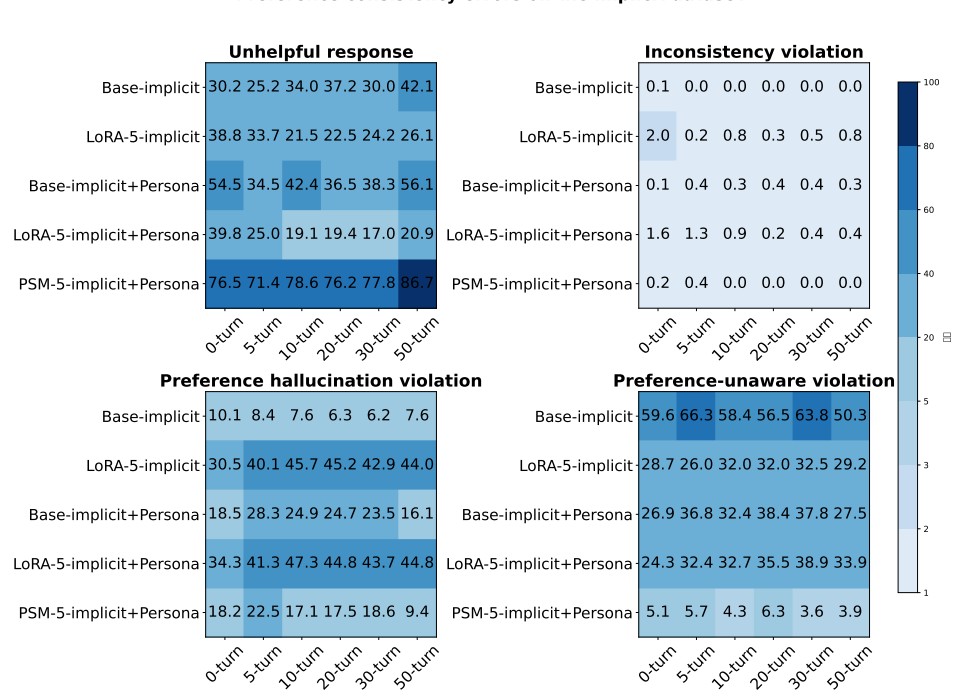

Figure 16: Preference consistency error types under different numbers of inserted irrelevant turns (n-turn).

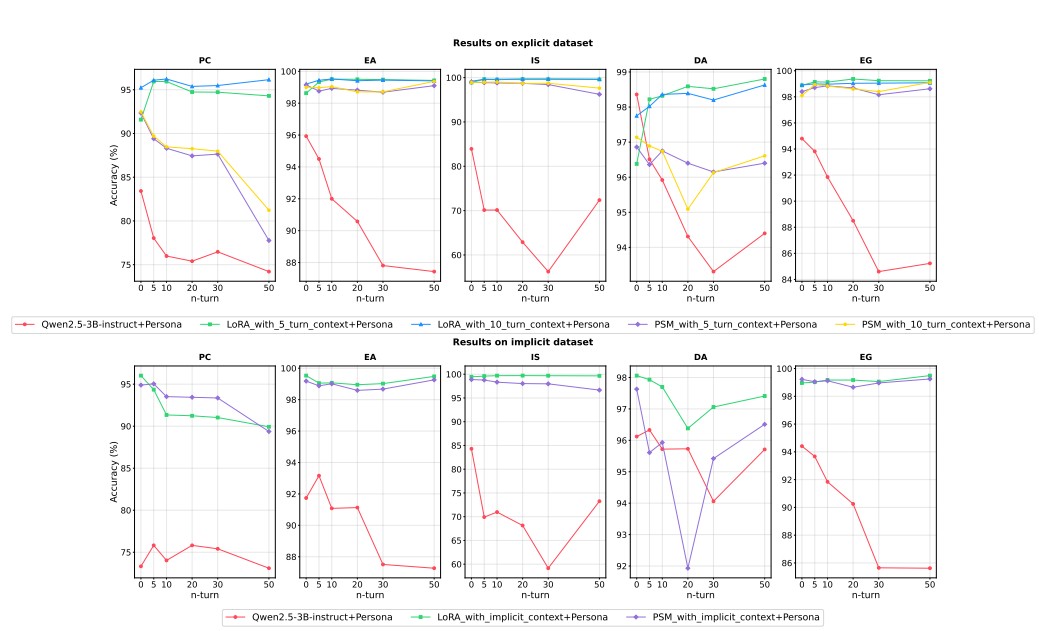

Figure 17: Accuracy of LLMs on preference consistency (PC) and child-oriented dimensions under different numbers of inserted irrelevant turns (n-turn).

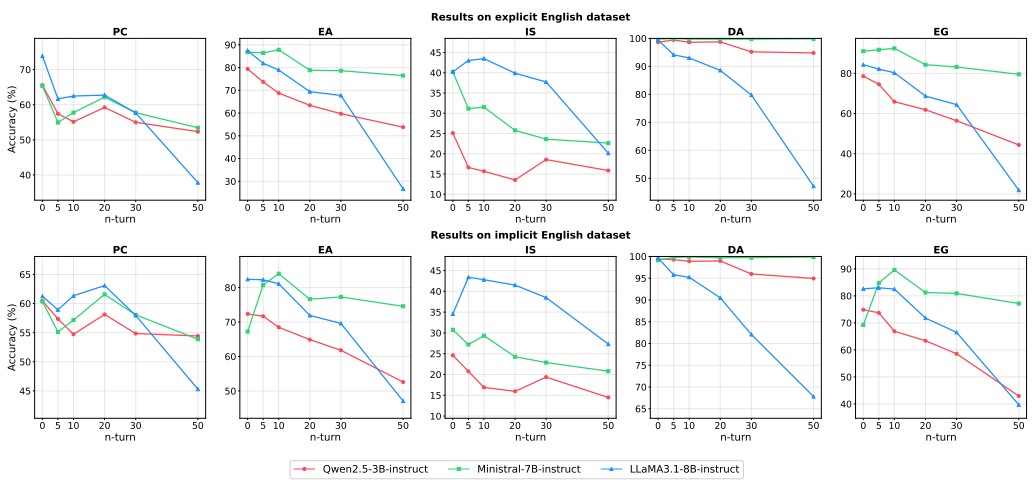

Figure 18: LLMs performances on preference consistency (PC) and the child-oriented evaluation under different numbers of inserted irrelevant dialogue turns on the English dataset.

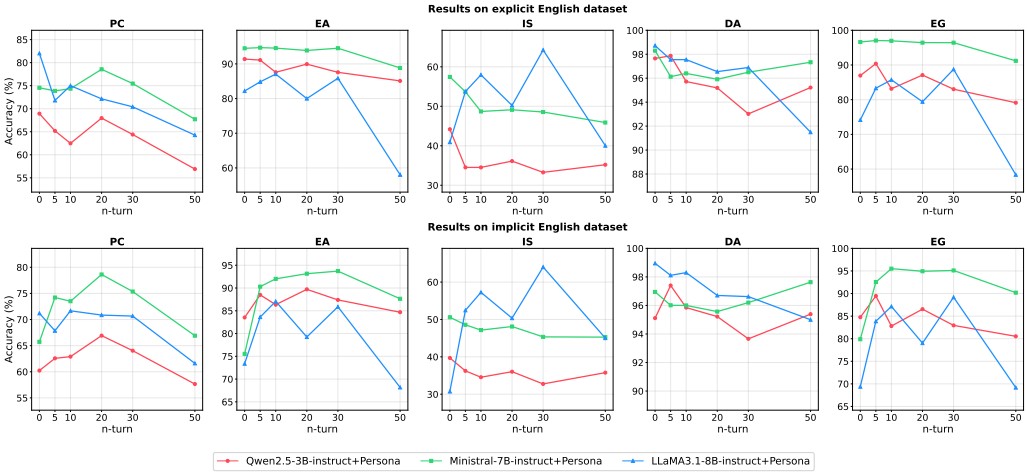

Figure 19: LLMs performances on preference consistency (PC) and the child-oriented evaluation under different numbers of inserted irrelevant dialogue turns on the English dataset, after integrating persona information into the prompt.

