# OpenReview forum: "ChildEval: How Large language models meet children’s personalities"
_ICLR.cc/2026/Conference — ICLR 2026 Conference Withdrawn Submission_

### Official Review · Reviewer_meKb · 2025-10-25

**Soundness:** 2
**Presentation:** 3
**Contribution:** 2
**Rating:** 4
**Confidence:** 3

**Summary:**

The authors introduce ChildEval - an evaluation benchmark for LLMs to rate their response from the perspective of children (3-6 years old).
They build a synthetic dataset of child-eval conversations + metrics to evaluate how well a LLM responds to a child.
Those metrics are based on custom prompts - showing a QWEN model the history, LLM_response and asking to give yes/no related to the criterion to evaluate.

Using this dataset and metrics, they evaluate 5 state of the art open-weight LLMs to see how good they are at responding to children. They show that LLMs struggle to keep their response consistent with child preferences after long and implicit conversations.They show that finetuning those LLMs on ChildEval improves evaluation performance.

**Strengths:**

Strengths:
- The paper introduces a synthetic dataset that looks soundly constructed for child-LLM dialogue, which can be reused by other researchers focusing on child LLM interaction. This is a novel domain that hasn't been explored.
- The paper introduces evals for child-LLM interaction - which is great for a new field and can be used as a benchmark for following papers.

**Weaknesses:**

- It is not clear in this paper how LORA finetuning was done to improve performance : there is no mention of train/test splits and it is unclear if LORA fine-tuning was done on the same data that is used for evaluations.
- The paper constructs personas and preferences as separate but some of the preferences are actually already included in personas. This make the finding "preference consistency increases when adding persona to the prompt" weaker.
- The 4 evals that are introduced are based on a prompt + response for QWEN - and numbers are then taken as ground truth for model comparison. It would be good to validate the stability of those metrics by comparing metric stability across different judge models.
- There is no correlation between the 4 metrics that are introduced and any real data/real human preference. The dataset is from synthetic data and the evals are from a synthetic Judge.

**Questions:**

What was the methodology used for LORA fine-tuning (training data, evaluation data)?
How strongly do the introduced metrics correlate with actual human preference?

---

> ### Author Response · Authors · 2025-11-20
>
> Dear Reviewer meKb:
>
> We sincerely appreciate your review and recognition of our paper’s strengths! Please see below for the point-to-point response:
>
> **Response to Weakness 1 (It is not clear in this paper how LORA finetuning was done to improve performance : there is no mention of train/test splits and it is unclear if LORA fine-tuning was done on the same data that is used for evaluations.".):**
>
> Thank you for your insightful comments, and we apologize for not making this point sufficiently clear in the original version. We generated 46K samples and randomly split them into training and test sets with an 8:2 ratio. The test data was strictly held out and never used during LoRA fine-tuning. We will include a detailed description of this data split and the fine-tuning procedure in the revised manuscript.
>
>
> **Response to Weakness 2 (The paper constructs personas and preferences as separate but some of the preferences are actually already included in personas. This make the finding "preference consistency increases when adding persona to the prompt" weaker.):**
>
> Thank you for your thoughtful comment. We agree that some preference information may overlap with the static persona descriptions. However, static personas are, by design, limited in their ability to capture the dynamic and situational nature of individual preferences. In contrast, the dialogue-level preferences are intended to represent dynamic traits that can vary across contexts. These two components may align, differ, or even conflict with each other. In our data generation process, we did not enforce any constraint requiring them to be consistent. This design choice reflects realistic conversational scenarios, where a person’s stable persona and moment-to-moment preferences are not always perfectly aligned. Therefore, the observed improvement in preference consistency when adding persona information remains meaningful.
>
>
> **Response to Weakness 3 (The 4 evals that are introduced are based on a prompt + response for QWEN - and numbers are then taken as ground truth for model comparison. It would be good to validate the stability of those metrics by comparing metric stability across different judge models.):**
>
> Thank you for your valuable comment. To validate the stability of the proposed metrics, we randomly sampled 100 examples covering all preference types and evaluated them using DeepSeek-R1. We then calculated the agreement rates between DeepSeek-R1 and Qwen2.5-72B. In addition, we computed the agreement rates of both DeepSeek-R1 and Qwen2.5-72B separately against manual human checks. The results are summarized in the tables below:
> ### Preference consistency(which adopts the established evaluation criteria (Zhao et al.,2025a))
> | Error Type                | Qwen2.5-72B vs Human | DeepSeek-R1 vs Human | DeepSeek-R1 vs Qwen2.5-72B |
> |----------------------------|---------------|--------------------|-------------------|
> | Acknowledge Preference?    | 83%           | 78%                | 76%               |
> | Violate Preference?        | 88%           | 89%                | 81%               |
> | Hallucinate Preference?    | 72%           | 64%                | 70%               |
> | Helpful Response?          | 99%           | 96%                | 97%               |
> | Preference Consistency     | 87%           | 86%                | 79%               |
>
> ### Child-Oriented Evaluation
>
> | Error Type                         | Qwen2.5-72B vs Human | DeepSeek-R1 vs Human | DeepSeek-R1 vs Qwen2.5-72B |
> |------------------------------------|---------------|--------------------|-------------------|
> | Emotional Adaptation (EA)           | 99%           | 97%                | 98%               |
> | Interaction Scaffolding (IS)        | 100%          | 99%               | 99%               |
> | Developmental Appropriateness (DA)  | 100%          | 98%               | 98%              |
> | Engagement (EG)                     | 100%          | 99%               | 99%               |
>
> In terms of preference consistency, the agreement rate between Qwen and human judgments is 87%, while DeepSeek achieves 86% agreement with human judgments. The agreement rate between Qwen and DeepSeek is 79%. These results demonstrate that the model evaluations are highly consistent, and both models’ outcomes align well with human evaluations, supporting the reliability of our metrics across different judge models. Furthermore, for evaluations targeting children, the results of the two models show even higher consistency with each other and with human judgments, indicating robust evaluation performance in child-focused settings.

---

> ### Author Response · Authors · 2025-11-20
>
> **Response to Weakness 4 (There is no correlation between the 4 metrics that are introduced and any real data/real human preference. The dataset is from synthetic data and the evals are from a synthetic Judge.):**
>
>
> We thank the reviewer for the valuable feedback. Collecting real-world data on children’s preferences is extremely difficult. Our evaluation uses synthetic data and a synthetic judge, but human verification showed high agreement, demonstrating that the protocol is reliable. It provides a controlled and evaluable framework for studying personalization and model behavior, enabling systematic experiments under safety and behavioral constraints. This framework serves as a first step toward more comprehensive modeling approaches in future work.
>
> **Response to Question 1 (What was the methodology used for LORA fine-tuning (training data, evaluation data)? How strongly do the introduced metrics correlate with actual human preference?):**
>
> Thank you for your question.For LoRA fine-tuning, we used fully synthetic data, which was split into training and evaluation sets in an 8:2 ratio. Each training sample consists of a preference specification, a long irrelevant dialogue context, a preference-related utterance, and the corresponding response:
>
> **[Explicit Preference/Implicit] + [Irrelevant Context] + [Preference-related Utterance] + [Response]**
>
> Specially, for training data
>
> The **[Irrelevant Context]** was randomly sampled from the WildChat-1M dataset, with variable context lengths.
>
> For testing data
>
> The **[Irrelevant Context]** was also sampled from WildChat-1M, but all evaluation samples used  a fixed context length. This design ensures a controlled evaluation environment, allowing us to reliably measure the model’s preference adherence while eliminating the influence of context length variations.
>
> The long context was included for two main reasons: first, the base model already demonstrates strong adherence to preferences under short-context conditions; second, it helps simulate realistic dialogue scenarios, allowing the model to generate preference-consistent responses even in complex conversations. **During training, the model only learns to generate the Response, while the preceding context and preference serve as conditional input to guide generation**.
>
> Regarding correlation with actual human preferences, we acknowledge that the evaluation metrics are based on synthetic data rather than direct measurements from real children. However, as discussed in our "Response to Weakness 3", these metrics provide a reliable and reproducible proxy for children preference adherence, enabling meaningful comparison between models in a controlled setting. While they do not replace real human data, they offer a practical framework for evaluating child-oriented personalization, which can be further calibrated with ethically collected real data in future work.

---

### Official Review · Reviewer_9AiT · 2025-10-26

**Soundness:** 2
**Presentation:** 3
**Contribution:** 2
**Rating:** 4
**Confidence:** 4

**Summary:**

This paper introduces ChildEval, a benchmark to evaluate how well Large Language Models can understand and respond to children's (ages 3-6) preferences in conversational settings. The benchmark comprises 29K synthetically generated children's persona profiles created through a three-step pipeline: 1) Persona Generation: Used Qwen2.5-72B to generate diverse child personas through iterative generation-and-refinement, with FAISS-based filtering to remove semantically similar profiles; 2) Preference Generation: Created 46K preferences from personas and sampled topics, expressed as first-person statements that can be revealed either explicitly or implicitly; 3) Dialogue Construction: Generated child-LLM conversations that naturally embed implicit preferences using prompt-based generation.

The benchmark introduces fine-grained child-centric metrics beyond standard Preference Consistency (PC):
- Emotional Adaptation (EA): Sensitivity to children's emotions
- Interaction Scaffolding (IS): Guiding participation with questions/hints
- Developmental Appropriateness (DA): Age-appropriate language and complexity
- Engagement (EG): Creating lively, interesting interactions

Experiments evaluated 5 state-of-the-art open-source LLMs (Qwen2.5-3B, Qwen3-4B, LLaMA3.1-8B, DeepSeek-R1, Mistral-7B). Results showed LLMs struggle with long-term personalization and implicit preference inference. Both LoRA and PSM fine-tuning methods significantly improve performance over base models on both preference consistency and child-oriented evaluation.

**Strengths:**

The work studies an important research direction: formulating child-LLM interaction as a personalization problem, addressing an important and underexplored application area with significant societal impact, and offers essential resources:
- The first large-scale benchmark with child-specific evaluation dimensions (46K preferences across 14 developmental topics)
- Novel child-centric evaluation framework beyond generic preference consistency: Emotional Adaptation (EA), Interaction Scaffolding (IS), Developmental Appropriateness (DA), and Engagement (EG)
- Comprehensive problem formulation distinguishing explicit vs. implicit preferences in long-context multi-session dialogues.
- Experimental analysis across 5 state-of-the-art LLMs and 3 adaptation strategies (prompting, LoRA, PSM), revealing that: LLMs struggle significantly with implicit preference inference and long-term personalization.

**Weaknesses:**

The benchmark does not provide sufficient details on data quality verification or comprehensive evaluation validation. Here are the key limitations:

Dataset quality:
- The training and evaluation set are entirely generated by Qwen2.5-72B with no validation against actual child-LLM conversations. It is unclear if generated dialogues match authentic child speech/text patterns (word omissions, semantic errors).
- No user studies with actual children or domain experts to verify realistic scenarios.
- The dataset might contain model-specific biases given it is entirely generated by Qwen2.5-72B, then evaluated using Qwen2.5-72B-Instruct, which might explain why Qwen3-4B-instruct outperform DeepSeek-R1 in Table 2.

Evaluation Validation:
- No human evaluation to validate LLM-as-judge approach. The LLM-judge prompt for EA and DA does not provide the evaluation guideline for different age (e.g., 3 year-old vs 6 year-old).
- All metrics evaluated by Qwen2.5-72B-Instruct - single model bias
- No error analysis on evaluation - unclear how often the evaluator makes mistakes

**Questions:**

- What are the data quality filters applied other than  FAISS-based similarity filtering? What types of errors were caught during data cleaning?

---

> ### Author Response · Authors · 2025-11-20
>
> Dear Reviewer 9AiT:
>
> We sincerely appreciate your review and recognition of our paper’s strengths! Please see below for the point-to-point response:
>
> **Response to Weakness 1 (The training and evaluation set are entirely generated by Qwen2.5-72B with no validation against actual child-LLM conversations. It is unclear if generated dialogues match authentic child speech/text patterns (word omissions, semantic errors".):**
>
> We thank the reviewer for raising this concern.To partially validate the synthetic data, our team manually inspected a subset of samples and referred to information from two volunteers with children aged 3–6 who had interacted with LLMs to confirm plausibility.
>
> We acknowledge that synthetic data cannot fully capture real child speech (e.g., omissions), but such variations have limited impact on the core personalization signals, as children typically express preferences in simple, direct ways. Abstracting these details enables a controlled,reproducible framework focused on preference-related reasoning.
>
> Since real child–LLM data are difficult to obtain for ethical, privacy, and annotation reasons, our goal is not to replace real data but to provide a standardized, extensible framework that can later be calibrated with ethically collected child data.
>
> **Response to Weakness 2 (No user studies with actual children or domain experts to verify realistic scenarios.):**
>
> We thank the reviewer for this comment. We acknowledge that no user studies with actual children or domain experts were conducted due to ethical, privacy, and practical constraints. To partially address this, our team manually inspected a subset of samples and referred to information from two volunteers with children aged 3–6 who had interacted with LLMs, as well as publicly available cases, to verify the plausibility of our data.
>
> Because real child–LLM interactions are difficult to obtain, our work serves as an initial step toward child-oriented evaluation, offering a foundation for future studies that may incorporate ethically collected real data. Our goal is not to replace real data, but to provide a standardized and extensible framework that can be calibrated when such data become available.
>
> **Response to Weakness 3 (The dataset might contain model-specific biases given it is entirely generated by Qwen2.5-72B, then evaluated using Qwen2.5-72B-Instruct, which might explain why Qwen3-4B-instruct outperform DeepSeek-R1 in Table 2..):**
>
> We thank the reviewer for raising this important concern. We note that the evaluation conditions for Qwen3-4B and DeepSeek-R1 differ: Qwen3-4B was deployed locally, whereas DeepSeek-R1, due to its size, was evaluated via API. We will clarify this distinction in the revised manuscript.
>
> As shown in Figure 3, DeepSeek-R1 performs better on preference consistency—reflecting more deliberate, preference-aware reasoning—while Qwen3-4B performs better on child-oriented evaluations, likely due to better alignment with the spontaneous and varied nature of child-like speech. These differences reflect task alignment rather than evaluation bias.
>
> To further assess potential model-specific bias, we manually inspected 100 randomly sampled examples across all preference types using DeepSeek-R1. We computed **agreement rates** between DeepSeek-R1 and Qwen2.5-72B, and also compared each model’s outputs with human judgments. As shown in the tables below, the evaluations from both models are highly consistent and closely match human assessments. Notably, for child-targeted evaluations, agreement was even higher, indicating robust and reliable evaluation performance in this setting.
>
> ### Preference consistency(which adopts the established evaluation criteria (Zhao et al.,2025a))
> | Error Type                | Qwen2.5-72B vs Human | DeepSeek-R1 vs Human | DeepSeek-R1 vs Qwen2.5-72B |
> |----------------------------|---------------|--------------------|-------------------|
> | Acknowledge Preference?    | 83%           | 78%                | 76%               |
> | Violate Preference?        | 88%           | 89%                | 81%               |
> | Hallucinate Preference?    | 72%           | 64%                | 70%               |
> | Helpful Response?          | 99%           | 96%                | 97%               |
> | Preference Consistency     | 87%           | 86%                | 79%               |
>
> ###   Child-Oriented Evaluation
> | Error Type                         | Qwen2.5-72B vs Human | DeepSeek-R1 vs Human | DeepSeek-R1 vs Qwen2.5-72B |
> |------------------------------------|---------------|--------------------|-------------------|
> | Emotional Adaptation (EA)           | 99%           | 97%                | 98%               |
> | Interaction Scaffolding (IS)        | 100%          | 99%               | 99%               |
> | Developmental Appropriateness (DA)  | 100%          | 98%               | 98%              |
> | Engagement (EG)                     | 100%          | 99%               | 99%               |

---

> > ### Author Response · Authors · 2025-11-20
> >
> > **Response to Weakness 4 (No human evaluation to validate LLM-as-judge approach. The LLM-judge prompt for EA and DA does not provide the evaluation guideline for different age (e.g., 3 year-old vs 6 year-old)..):**
> >
> >
> > We thank the reviewer for highlighting the lack of age-specific distinctions in our LLM-judge prompts. This is a valid point. Our approach evaluated the 3-6 year-old range as a single group, primarily because the age was part of the persona, and we aimed to maintain consistency across experimental conditions (including those without the persona). We acknowledge that this limits the granularity of our analysis. We will clarify this methodological trade-off in the paper and explore more precise, age-sensitive evaluation frameworks in future work.
> >
> > **Response to Question 1 (What are the data quality filters applied other than FAISS-based similarity filtering? What types of errors were caught during data cleaning?):**
> >
> > **Human review**: Team members manually checked the generated dialogues to ensure alignment with the intended child–robot interaction scenarios.
> >
> > **Prompt refinement and stricter role enforcement**:
> > * Role confusion: The LLM occasionally confused the roles of child and robot—for instance, a child addressing the robot as “Mama,” or the robot exhibiting human-like behaviors inconsistent with its role. We introduced stricter role definitions in prompts to reduce such errors.
> > * Child-like characteristics: Some outputs still felt “machine-like” and lacked natural child traits. To mitigate this, we incorporated child-specific linguistic cues, such as interjections (“ah,” “ne”), though some residual artificiality remains.
> > * Instruction adherence: During dialogue generation, the model sometimes forgot key requirements from the prompt (e.g., persona traits or interaction rules). To address this, we explicitly instructed the model to check and follow these reinforced elements in its output, and introduced a self-check step to verify that the output conforms to the core prompt requirements.
> >
> > **Diversity considerations**: Although deduplication was performed, we observed that personas with high embedding similarity could still represent distinct characters (e.g., both labeled “curious,” but one a 4-year-old who loves dancing, the other a 6-year-old who loves singing). Setting similarity thresholds too high risks removing meaningful diversity, while thresholds too low fail to control redundancy. We iteratively tuned these parameters to balance diversity and consistency.
> >
> > These combined measures help reduce common errors and improve role fidelity, child-likeness, and diversity, enhancing the dataset's overall reliability for downstream evaluation. Importantly, the goal of this synthetic dataset is not to replace real child data, but to provide a controllable framework and evaluation platform for assessing models under personalized and safety-constrained scenarios.

---

### Official Review · Reviewer_M6CE · 2025-10-31

**Soundness:** 2
**Presentation:** 2
**Contribution:** 2
**Rating:** 2
**Confidence:** 4

**Summary:**

The study proposes ChildEval to evaluate LLMs' personalization capability toward preschool children (ages 3-6). The benchmark features 29K synthetic child persona profiles, with both explicit and implicit preferences embedded within 6 to 10-turn dialogues. The evaluation protocol goes beyond traditional Preference Consistency (PC), introducing child-specific metrics: Emotional Adaptation (EA), Interaction Scaffolding (IS), Developmental Appropriateness (DA), and Engagement (EG).

**Strengths:**

The paper addresses LLM interaction and personalization for children, a critical and underexplored area given LLMs' increasing role as a core medium for human-computer interaction.

**Weaknesses:**

- Neglect of Safety and Mental Health (Core Concern): The paper focuses too narrowly on preference matching and completely omits the crucial dimensions of LLM security, risk mitigation, and the ability to guide children toward healthy mental development. This is considered a more essential scenario and an unacceptable omission in any LLM-child interaction assessment.

- Flawed Personality Modeling: The methodology relies on generating personas via LLM-created "explicit static persona descriptions." This approach lacks discussion on what "child personality" truly is, risking significant "personality bias" and failing to capture the implicit, dynamic, and complex nature of a child's true personality.

- Data Reliability and Ethical Issues: The data is entirely model-driven/synthetic with no human participation. This raises serious doubts about the reliability and validity of the fabricated personalities, questioning whether they reflect the true distribution of human/child personalities.

- Lack of Interdisciplinary Rigor: The work is perceived as irresponsibly applying a general-purpose personality modeling framework to a sensitive child scenario without sufficient consideration of social, psychological, or educational complexities. This purely technical approach is viewed as lacking practical meaning and failing to align with the ultimate goal of ensuring LLMs serve a child's healthy growth.

**Questions:**

1. Why does the smaller Qwen3-4B model significantly outperform much larger models like DeepSeek-R1 in Table 2? This conclusion violates the general trend observed in the scaling law. What is the authors' explanation for this anomalous phenomenon?

2.  Why are the results for DeepSeek-R1 in Figure 2 only presented up to 10 dialogue turns, while other models (such as Qwen3-4B and Llama2-7B) are evaluated over longer contexts (e.g., 20 turns)?

**Details Of Ethics Concerns:**

The exclusive reliance on LLMs to synthesize child persona data without human validation poses a critical ethical risk. Given the sensitivity and complexity of child personalities, the model-generated data is highly likely to contain significant bias and inaccurate stereotypes, lacking necessary psychological rigor. Evaluating or training models on this biased data can lead to misleading or harmful 'personalized' interactions, potentially impeding a child's healthy cognitive and emotional development. This approach compromises the benchmark’s credibility and violates safety standards for research involving vulnerable populations.

---

> ### Author Response · Authors · 2025-11-20
>
> Dear Reviewer M6CE:
>
> We sincerely appreciate your review and recognition of our paper’s strengths! Please see below for the point-to-point response:
>
> **Response to Weakness 1 (Neglect of Safety and Mental Health (Core Concern): The paper focuses too narrowly on preference matching and completely omits the crucial dimensions of LLM security, risk mitigation, and the ability to guide children toward healthy mental development. This is considered a more essential scenario and an unacceptable omission in any LLM-child interaction assessment.".):**
>
> Thank you for raising this important concern. We fully recognize that ensuring child safety is both essential and highly challenging.
>
> Our work builds on a safety-aligned base model, and the initial construction of child profiles involved careful manual selection and review to ensure that the simulated children remain healthy, positive, and age-appropriate, while preserving a small set of individual preferences. As the reviewers note, children are a special population, and safety involves far more than avoiding harmful outputs. In our paper, we emphasize that understanding and mitigating risks in real child–LLM interactions is a key direction for our future work, involving issues such as privacy protection, bias reduction, and potential cognitive or emotional impacts.
>
> Because collecting real-world child data is difficult and sensitive, our goal is not to replace authentic child interactions, but to provide a controlled and reproducible platform for preliminary analysis of personalized and safety-related behaviors. To further ensure data safety and suitability, all benchmark samples were manually reviewed, including anonymization of names, removal of inappropriate content, and checks for potential safety risks.
>
>
>
> **Response to Weakness 2 (Flawed Personality Modeling: The methodology relies on generating personas via LLM-created "explicit static persona descriptions." This approach lacks discussion on what "child personality" truly is, risking significant "personality bias" and failing to capture the implicit, dynamic, and complex nature of a child's true personality..):**
>
> We thank the reviewer for pointing this out. Indeed, personality is a complex and multifaceted construct, and the static "persona" descriptions generated by LLMs represent only a simplified view of a child’s traits:such as age, gender, and relatively stable preferences or characteristics (e.g., milk allergy). True child personality is dynamic, context-dependent, and can manifest through multiple channels, including behavior patterns in apps or daily interactions, which may serve as implicit indicators of preferences and tendencies. Although we aim to better understand child personality, no existing work claims to fully capture its richness. Our work does not attempt to fully capture the richness of a child’s personality or to replace real child data. Rather, it provides a controlled and evaluable framework for studying personalization and model behavior, enabling systematic experiments under safety and behavioral constraints. This framework can serve as a first step toward more comprehensive modeling approaches in future work.
>
>
> **Response to Weakness 3 (Data Reliability and Ethical Issues: The data is entirely model-driven/synthetic with no human participation. This raises serious doubts about the reliability and validity of the fabricated personalities, questioning whether they reflect the true distribution of human/child personalities.):**
>
> We thank the reviewer for raising this important point. While it is true that the benchmark data is primarily generated by LLMs, we took several steps to enhance reliability and ethical compliance. First, the initial child profiles were manually crafted and reviewed to ensure they reflect healthy and positive characteristics, while retaining some individual interests. Second, all generated dialogues underwent manual review, including anonymization of personal names, removal of harmful content, content-level checks, and the elimination of any latent manipulative guidance. These procedures help reduce risks related to privacy, harmful content, and ethical concerns.
>
> We emphasize that although we aim to better understand child personality,the synthetic data is not intended to replace real child data. Rather, it provides a controlled and reproducible experimental platform for evaluating model behavior under personalization and safety constraints. The framework allows systematic study of model robustness and personalized responses, while respecting ethical boundaries and minimizing risks associated with collecting real child data.

---

> > ### Author Response · Authors · 2025-11-20
> >
> > **Response to Weakness 4 (Lack of Interdisciplinary Rigor: The work is perceived as irresponsibly applying a general-purpose personality modeling framework to a sensitive child scenario without sufficient consideration of social, psychological, or educational complexities. This purely technical approach is viewed as lacking practical meaning and failing to align with the ultimate goal of ensuring LLMs serve a child's healthy growth..):**
> >
> >
> > We thank the reviewer for raising this important concern. We acknowledge that child development is inherently complex and interdisciplinary, encompassing social, psychological, and educational dimensions. Our work does not claim to fully model or intervene in a child’s growth, nor does it attempt to replace expertise from developmental psychology or education. Instead, our goal is to provide a controlled and reproducible experimental framework for evaluating LLM behavior under personalization and safety constraints. This framework allows systematic study of technical approaches—such as persona-based modeling and preference alignment—while adhering to ethical safeguards, supporting preliminary insights that may inform future interdisciplinary research.
> >
> >
> > **Response to Question 1 (Why does the smaller Qwen3-4B model significantly outperform much larger models like DeepSeek-R1 in Table 2? This conclusion violates the general trend observed in the scaling law. What is the authors' explanation for this anomalous phenomenon?):**
> >
> > We thank the reviewer for this insightful question. We note that the evaluation conditions for Qwen3-4B and DeepSeek-R1 differ, which may partly explain the observed performance discrepancy. Specifically, Qwen3-4B was downloaded and deployed locally, allowing full control over inference settings and batch processing. In contrast, DeepSeek-R1, due to its large parameter size, was accessed via API, which may introduce latency, rate limits, or other deployment constraints that could affect its performance. We will clarify these differences in the revised manuscript to avoid potential misunderstanding.
> >
> > Furthermore, even if deployment differences are not the primary factor, the lower performance of DeepSeek-R1 on the child-oriented dataset does not imply a violation of scaling laws. Scaling laws describe general trends across model size and performance, typically observed under large-scale data and broad tasks. They do not guarantee superior performance in every scenario. In tasks with small datasets or specific stylistic requirements, smaller models can outperform larger ones—not because scaling laws are contradicted, but because the smaller models are better suited to the particular conditions of the task.
> >
> > Notably, in our paper, DeepSeek-R1 shows much stronger performance on preference consistency, a metric associated with more thoughtful, preference-aware reasoning. In contrast, Qwen3-4B performs better on the child-oriented evaluations, likely because it more naturally captures the spontaneous and variable characteristics of child-like speech. These observations highlight task alignment differences, rather than any contradiction of scaling trends.
> >
> > **Response to Question 2 (Why are the results for DeepSeek-R1 in Figure 2 only presented up to 10 dialogue turns, while other models (such as Qwen3-4B and Llama2-7B) are evaluated over longer contexts (e.g., 20 turns)?):**
> >
> > Thank you for raising this question. As noted earlier, when exceeding 20 turns, the API call for DeepSeek-R1 surpasses its maximum context window of 8192 tokens, which results in the request being rejected. Consequently, the evaluation could only include up to 10 dialogue turns for DeepSeek-R1, and this limitation arises from the API itself rather than from any experimental restriction on our side.

---

### Official Review · Reviewer_pEdu · 2025-11-01

**Soundness:** 3
**Presentation:** 3
**Contribution:** 2
**Rating:** 4
**Confidence:** 4

**Summary:**

This paper introduces a new benchmark called ChildEval, designed to assess the ability of large language models (LLMs) to understand, reason, and follow children's preferences in long-context conversations. The authors note that existing personalized research primarily focuses on adults, while children exhibit significantly different interaction patterns and needs, lacking corresponding evaluation metrics. To address this gap, ChildEval incorporates 29,000 synthetic child personas, each associated with explicit and implicit preferences in daily life and developmental contexts.

Beyond the dataset itself, this work proposes a fine-grained evaluation protocol tailored for children. It introduces four new child-oriented assessment dimensions alongside the traditional “Preference Consistency (PC)”: Emotional Adaptation (EA), Interaction Scaffolding (IS), Developmental Appropriateness (DA), and Engagement (EG). .

In the experimental section, the authors evaluated multiple open-source LLMs, verifying the impact of different personalization representation methods (such as incorporating personas into prompts) on model performance. They demonstrated that fine-tuning on ChildEval enhances models' ability to personalize for children. Experimental results reveal challenges existing LLMs face in handling children's implicit preferences and long dialogue histories, emphasizing the importance of comprehensive evaluation.

**Strengths:**

- ChildEval is thoughtfully designed. It not only incorporates children's basic profiles but also distinguishes between explicit and implicit preferences, with the latter posing greater demands on the model's reasoning capabilities. By inserting irrelevant dialogues into conversation histories to simulate long-context scenarios, the evaluation becomes more realistic and challenging.
- One of the key contributions of this paper is the introduction of four new evaluation dimensions tailored for children: EA, IS, DA, and EG. These dimensions go beyond merely assessing whether responses align with children's preferences, instead evaluating LLM responses across multiple dimensions—interaction quality, emotional support, educational value, and fun—all of which are crucial for child users. This evaluation framework provides a valuable foundation for future research in this area.

**Weaknesses:**

- The entire ChildEval benchmark (including child profiles and dialogues) was generated by LLM. Although the authors employed methods like deduplication to ensure diversity, synthetic data may not fully capture the complexity, creativity, and unpredictability of authentic child language. Biases inherent in the generative model itself may also be introduced into the dataset. The paper's description of the “self-verification” process is insufficiently clear, lacking human validation steps to ensure the quality and authenticity of the generated data. This represents a major limitation of the work.
- The paper employs LLM to evaluate four dimensions of child-oriented content. While “LLM-as-a-judge” is a prevalent evaluation approach today, its reliability remains under investigation. Conducting human evaluations on a small subset of data and calculating consistency with LLM assessments would significantly enhance the credibility of evaluation results.

**Questions:**

- Could you provide more details about the “self-verification” mechanism during dialogue generation? When constructing the dataset, was any form of manual sampling evaluation or qualitative analysis conducted to validate the authenticity and quality of the generated child profiles and dialogues—particularly those used to infer implicit preferences?
- When using LLM as an evaluator, have you considered or conducted any small-scale human evaluations to validate the accuracy of the LLM's judgments? How do you perceive and address potential biases that may arise from using an LLM to evaluate other LLMs?
- Experimental results indicate that the LoRA method generally outperforms your proposed PSM in preference consistency. Could you elaborate further on potential scenarios or aspects where PSM might hold advantages over LoRA? For instance, does it exhibit particular strengths in computational efficiency, deployment convenience, or interpretability?

---

> ### Author Response · Authors · 2025-11-20
>
> Dear Reviewer pEdu:
>
> We sincerely appreciate your review and recognition of our paper’s strengths! Please see below for the point-to-point response:
>
> **Response to Weakness 1 (The entire ChildEval benchmark (including child profiles and dialogues) was generated by LLM. Although the authors employed methods like deduplication to ensure diversity, synthetic data may not fully capture the complexity, creativity, and unpredictability of authentic child language. Biases inherent in the generative model itself may also be introduced into the dataset. The paper's description of the “self-verification” process is insufficiently clear, lacking human validation steps to ensure the quality and authenticity of the generated data. This represents a major limitation of the work.".):**
>
>
> We appreciate the reviewer’s comments regarding potential biases and the clarity of the “self-verification” process. During data construction, we used human-in-the-loop methods for generation and validation. We analyzed real interactions between 3–6-year-old children and LLMs, including samples collected ourselves and provided by two consenting volunteers. Based on these, our team manually created over 20 child prototypes with healthy traits and individual differences (e.g., picky eating, shyness, curiosity) as seeds for generating child profiles.
>
> Regarding the clarity of the self-verification process, we apologize for the initial lack of detail. The model performs automated checks on the form and format of its outputs, as shown in Appendix Fig. 9，which includes seven verification items. All generated profiles were then manually reviewed by our team to ensure safety and child-appropriate expression. This human verification step compensates for the limitations of synthetic data and is essential for ensuring benchmark quality.
>
> We also acknowledge that generating preference data inevitably introduces biases—for example, real children are not consistently goal-directed, whereas our simulations may amplify certain tendencies. These biases are intentional and serve as controlled experimental tools, allowing systematic evaluation under diverse and extreme preference scenarios. The simulations are not meant to replicate real child behavior, but to test model robustness in a controlled and reproducible way.
>
> **Response to Weakness 2 (The paper employs LLM to evaluate four dimensions of child-oriented content. While “LLM-as-a-judge” is a prevalent evaluation approach today, its reliability remains under investigation. Conducting human evaluations on a small subset of data and calculating consistency with LLM assessments would significantly enhance the credibility of evaluation results
> .):**
>
> We appreciate the reviewer's comment and acknowledge this limitation.
> A large-scale human evaluation was not conducted due to the cost. Nevertheless, we have implemented several measures to ensure basic reliability:
>
> **Partial human cross-check**: We analyzed real interactions between 3–6-year-old children and LLMs, including samples collected ourselves and provided by two consenting volunteers whose households included children aged 3–6 who had interacted with the LLMs, helping to confirm the reasonableness of our evaluation.
>
> **Large-scale robustness testing**: Considering the inherent generation variability of LLM outputs, we conducted evaluations on over 9,000 test instances for each experiment to reduce random fluctuations and enhance stability through large-sample statistics.
>
> **Human check Metric**: We agree that human verification remains important. So we manually validated 100 randomly sampled examples covering all preference types and calculated the human-LLM agreement rates..
> ### Preference consistency(which adopts the established evaluation criteria (Zhao et al.,2025a))
> | Error Type |  Agreement rate |
> |------------|---------|
> | Acknowledge Preference?    | 83%  |
> | Violate Preference?    | 88%   |
> | Hallucinate Preference?    | 72%  |
> | Helpful Response?    | 99%   |
> | Preference consistency    | 87%   |
>
> While not a substitute for human review, the 87% agreement indicates that LLM-based evaluation is trustworthy and effective for scalable preference-consistency assessment.
> ### Child-Oriented Evaluation
> | Error Type |  Agreement rate |
> |------------|---------|
> | Emotional Adaptation (EA) | 99% |
> | Interaction Scaffolding (IS)    | 100%   |
> |Developmental Appropriateness (DA)    | 100%  |
> | Engagement (EG)    | 100%   |
>
> These high agreement rates do not indicate perfect measurement precision; rather, they reflect that the child-oriented criteria are broad, interpretable, and consistently identifiable by evaluators. Importantly, this characteristic does not undermine the core contribution of our work. The goal of the child-oriented evaluation is not fine-grained discrimination, but to verify whether the model satisfies the fundamental requirements and principles of child-directed communication.

---

> > ### Author Response · Authors · 2025-11-20
> >
> > **Response to Question 1 (Could you provide more details about the “self-verification” mechanism during dialogue generation? When constructing the dataset, was any form of manual sampling evaluation or qualitative analysis conducted to validate the authenticity and quality of the generated child profiles and dialogues—particularly those used to infer implicit preferences?):**
> >
> > Thank you for this question. The specific checks performed are detailed in Appendix Fig 9 (“Prompt used for generating child–LLM dialogue to infer the implicit preference: Part 2-Self-check and Outputs”). These mainly rely on prompt-based self-verification by the model to ensure structural and stylistic validity, including the following seven aspects:
> >
> > * Name check: No personal names used, compliant.
> > * Closing phrase check: No closing phrase in the last turn, compliant.
> > * Tone check: Language is mild and natural, matching the style of 3-6-year-old children.
> > * Assistant behavior check: Assistant is not personified and contains no self-involvement in activities.
> > * Turn count check: Exactly {n} turns (i.e., {n} <user> and {n} <assistant>).
> > * Tag check: All tags spelled correctly and fully closed.
> > * Preference inference check: From the dialogue, the child's attitude toward “xxx” can naturally reveal the
> > explicit preference
> >
> >
> > **Response to Question 2 (When using LLM as an evaluator, have you considered or conducted any small-scale human evaluations to validate the accuracy of the LLM's judgments? How do you perceive and address potential biases that may arise from using an LLM to evaluate other LLMs?
> > ):**
> >
> > Thank you for the insightful question. we performed small pilot checks in which team members manually reviewed a subset of the LLM-generated evaluations. These checks showed that the evaluator model’s judgments were generally aligned with human intuition for clear cases, while ambiguity occasionally led to disagreements.
> >
> > Regarding potential biases when using an LLM to evaluate another LLM, we fully acknowledge this as a known limitation of LLM-based evaluation pipelines. To mitigate this issue, we used an evaluator model different from the models being evaluated, which helps reduce self-preference bias. Importantly, even if the evaluator model exhibits its own systematic biases, as long as such biases are applied consistently across all evaluated models, the relative comparison remains meaningful.
> >
> > **Response to Question 3 (Experimental results indicate that the LoRA method generally outperforms your proposed PSM in preference consistency. Could you elaborate further on potential scenarios or aspects where PSM might hold advantages over LoRA? For instance, does it exhibit particular strengths in computational efficiency, deployment convenience, or interpretability?):**
> >
> > We thank the reviewer for the insightful question regarding potential advantages of PSM over LoRA. While current experimental results indicate that LoRA generally achieves higher preference consistency, PSM was designed with complementary goals. Specifically, PSM is a plug-and-play component capable of integrating diverse sources of personalization beyond static persona descriptions, such as daily behavior patterns or app usage paths. For LLM-based robots, the applications accessed by children within the robot environment can provide additional signals of individual preferences.
> > PSM also emphasizes interpretability and modularity, allowing components to be analyzed or swapped more easily. Although there has not yet been sufficient experimental time to fully quantify these aspects, the design of PSM allows for a flexible framework to fuse multiple personalization signals and support downstream analysis. This design makes PSM particularly suitable for scenarios requiring transparent, controllable, and integrative personalization, even if absolute preference consistency may be slightly lower than LoRA in current metrics.
> > It should also be noted that PSM has not yet been implemented for inference on vllm. Currently, outputs are generated using standard generate methods, which means that, in terms of computational efficiency, PSM may not fully leverage vllm’s advantages. Future work could explore integrating PSM directly into vllm inference to further optimize efficiency.

---

> > > ### Comment · Reviewer_pEdu · 2025-11-27
> > >
> > > I thank the authors for their response and the additional human verification data. While these efforts are appreciated, I remain concerned about the validity of using a fully synthetic benchmark to model the complex and unpredictable nature of children's behavior. Therefore, I will maintain my original score.

---

### Official Review · Reviewer_v1hT · 2025-11-01

**Soundness:** 2
**Presentation:** 3
**Contribution:** 2
**Rating:** 4
**Confidence:** 3

**Summary:**

This paper introduces ChildEval, a novel benchmark designed to evaluate how well LLMs can personalize interactions for children (ages 3-6) by inferring and following their preferences in --possibly long-- conversational settings. The benchmark includes 29K persona profiles, each paired with 6-10 turn dialogues (check Q1 for clarification), covering 5 top-level and 14 sub-level topics related to children's daily lives and development (art, cognitive development, nutrition, language, social-emotional development), in both Chinese and English. The authors propose a comprehensive evaluation framework covering Preference Consistency (Alignment score) and more child-oriented metrics (e.g., Appropriateness). Finally, the authors benchmarked four well-known models (Qwen, LLaMA, DeepSeek, Mistral) on the proposed dataset using three strategies: prompting, LoRA fine-tuning, and a novel architecture called Persona Steer Module. The authors find that the performance of LLMs degrades as irrelevant dialogue turns increase. Adding persona information helps consistency, and fine-tuning the model improves performance with different trade-offs (LoRA vs. proposed approach).

**Strengths:**

- Comprehensive benchmark and with 29K diverse persona profiles and realistic scenarios and diverse topics, and bilingual.
- Novel evaluation framework, with 4 new child-specific dimensions (EA, IS, DA, EG)
- Comprehensive evaluation with many SOTA LLMs, different evaluation strategies (prompting, LoRA, Persona Steer Module), long-context evaluation (up to 21K tokens) and several ablation and error analysis.

**Weaknesses:**

- All personas and dialogues are LLM-generated, and there is no verification that synthetic data reflects actual child behavior.
Authenticity concerns: May not capture true child communication patterns (e.g., word omissions, semantic errors mentioned but not evidenced)
- Lack of human evaluation to verify the quality of the proposed evaluation framework and metric.

Minor
- No comparison with frontier models ( GPT-4, Claude, or Gemini), even on a subset would have being an interesting analysis.

**Questions:**

Q1: how many actual dialogues and turns have being generated?

---

> ### Author Response · Authors · 2025-11-20
>
> Dear Reviewer v1hT:
>
> We sincerely appreciate your review and recognition of our paper’s strengths! Please see below for the point-to-point response:
>
> **Response to Weakness 1 (All personas and dialogues are LLM-generated, and there is no verification that synthetic data reflects actual child behavior. Authenticity concerns: May not capture true child communication patterns (e.g., word omissions, semantic errors mentioned but not evidenced".):**
>
> We appreciate the reviewer's concern. We acknowledge that synthetic data cannot fully replicate natural child communication, including omissions or semantic errors. However, these developmental irregularities do not substantially affect the core personalization signals we evaluate—children typically express preferences in simple forms that remain intact even without non-adult grammar. Thus, the absence of such surface-level errors does not compromise the validity of assessing preference alignment.
>
> While linguistic errors may influence LLM behavior in specific cases, excluding them allows us to establish a controlled and reproducible baseline for evaluating personalized reasoning without noise-induced variability. Real child–LLM data remain extremely difficult to obtain due to ethical, privacy, and annotation constraints. Our goal is not to replace real data but to build a standardized, extensible framework that can be calibrated once small-scale, ethically collected child data become available.
>
> **Response to Weakness 2 (Lack of human evaluation to verify the quality of the proposed evaluation framework and metric.):**
>
> We appreciate the reviewer's comment and acknowledge this limitation.
> A large-scale human evaluation was not conducted, primarily due to the cost and the need for reproducibility. Nevertheless, we have implemented several measures to ensure basic reliability:
>
> **Partial human cross-check**: Team members manually inspected a subset of samples to verify whether the model outputs aligned with the expected responses under the evaluation prompts. In this process, consent was obtained from two volunteers whose households included children aged 3–6 who had interacted with the LLMs, helping to confirm the reasonableness of our evaluation.
>
> **Large-scale robustness testing**: Considering the inherent generation variability of LLM outputs, we conducted evaluations on over 9,000 test instances for each experiment to reduce random fluctuations and enhance stability through large-sample statistics.
>
> **Human check Metric**: We agree that human verification remains important. So we manually validated 100 randomly sampled examples covering all preference types and calculated the human-LLM agreement rates..
> ### \#Preference consistency(which adopts the established evaluation criteria (Zhao et al.,2025a))
>
> | Error Type |  Agreement rate |
> |------------|---------|
> | Acknowledge Preference?    | 83%  |
> | Violate Preference?    | 88%   |
> | Hallucinate Preference?    | 72%  |
> | Helpful Response?    | 99%   |
> | Preference consistency    | 87%   |
>
> While not a substitute for human review, the 87% agreement indicates that LLM-based evaluation is trustworthy and effective for scalable preference-consistency assessment.
>
> ### \#Child-Oriented Evaluation
>
> | Error Type |  Agreement rate |
> |------------|---------|
> | Emotional Adaptation (EA) | 99% |
> | Interaction Scaffolding (IS)    | 100%   |
> |Developmental Appropriateness (DA)    | 100%  |
> | Engagement (EG)    | 100%   |
>
> These high agreement rates do not indicate perfect measurement precision; rather, they reflect that the child-oriented criteria are broad, interpretable, and consistently identifiable by evaluators. Importantly, this characteristic does not undermine the core contribution of our work. The goal of the child-oriented evaluation is not fine-grained discrimination, but to verify whether the model satisfies the fundamental requirements and principles of child-directed communication.

---

> > ### Author Response · Authors · 2025-11-20
> >
> > **Response to Weakness 3 (No comparison with frontier models ( GPT-4, Claude, or Gemini), even on a subset would have being an interesting analysis.):**
> >
> > We appreciate this valuable comment.
> >
> > As mentioned, our focus was on ensuring evaluation stability and reproducibility. To reduce LLM output variability, each experiment used around 9,000 samples with extensive personalization trials. Although we acknowledge the value of benchmarking against frontier models,API costs make large-scale personalized evaluations infeasible. Our study therefore prioritizes an open, reproducible, model-agnostic evaluation framework, which can accommodate such comparisons in the future.
> >
> > **Response to Question 1 ( how many actual dialogues and turns have being generated?.):**
> > Thank you for your question.
> >
> > We generated 46,113 samples, each consisting of:
> >      a persona;
> >      a single instance of an explicit preference and its related utterance;
> >      a dialogue from which the explicit preference can be inferred.;
> >
> > Each dialogue contains 6-10 turns. We observed that dialogues longer than 10 turns tend to show much noticeable deviations from realistic child–LLM interactions and may negatively affect the coherence and interpretability of the generated outputs. Limiting dialogue length in this way allows us to maintain a balance between generating sufficiently rich interactions and preserving output quality, while keeping the evaluation reproducible and reliable.
> >
> > Here is an example from the samples, in this example there are 7 turns in the implicit_dialogue.
> > ```xml
> > <example>
> > <persona>A 4-year-old child with a strong interest in language and expression. He enjoys mimicking the tone and intonation of adults, and although he sometimes makes people laugh due to inaccurate pronunciation, this does not hinder his enjoyment of expressing himself. Both of his parents come from bilingual families, so he naturally learned two languages in his daily life and can use them for basic communication. He is curious about books and stories, enjoys listening to various interesting stories told by his parents, and tries to express them in his own words. He also likes to mimic animal sounds, especially those of kittens and puppies, and he will laugh happily whenever he hears these sounds. His sensitive personality makes him a bit shy when encountering strangers, but he quickly adapts to new environments and uses language and expression to connect with others.</persona>
> >
> > <explicit_preference>I don't like sticky things; it feels so uncomfortable when they get on my hands, especially when I'm playing outside.</explicit_preference>
> > <related_utterance>Mom said she's going to take me on a picnic this weekend, have you ever been on a picnic?</related_utterance>
> >
> > <implicit_dialogue>
> > {'role': 'child', 'content': "Mom said she's going to take me to the park this weekend, can I bring some snacks?"}
> > {'role': 'assistant', 'content': 'Of course you can! What yummy food do you want to bring?'}
> > {'role': 'child', 'content': "I want to bring some cookies and fruits, but I don't like sticky things; it feels uncomfortable when they get on my hands."}
> > {'role': 'assistant', 'content': "Got it, then let's bring some dry cookies and fruits so your hands won't get sticky. What kind of cookies do you like the most?"}
> > {'role': 'user', 'content': "I like chocolate cookies, they are sweet and don't make my hands sticky."}
> > {'role': 'assistant', 'content': 'Chocolate cookies sound delicious! What fruits do you plan to bring?'}
> > {'role': 'user', 'content': "I want to bring apples and bananas, but banana peels are a bit sticky, so I'll just bring apples."}
> > {'role': 'assistant', 'content': 'Apples are a great choice, they are clean and easy to eat. What else do you plan to bring to the park?'}
> > {'role': 'user', 'content': "I also want to bring a small mat, so I won't touch the wet grass when I sit down."}
> > {'role': 'assistant', 'content': 'A small mat is a good idea, so you can sit comfortably. What do you plan to do at the park?'}
> > {'role': 'user', 'content': 'I want to run and jump, and bark like a dog, woof! Woof woof!'}
> > {'role': 'assistant', 'content': 'Ha ha, just like a real dog! Do you mimic other animals too?'}
> > {'role': 'user', 'content': 'I can also mimic the sound of a kitten, meow~ Sometimes I even tell stories to myself.'}
> > {'role': 'assistant', 'content': "That's awesome! I bet your stories are very interesting. What story will you tell at the park?"}
> > </implicit_dialogue>
> > </example>

---

### Note · Authors · 2025-12-26

I have read and agree with the venue's withdrawal policy on behalf of myself and my co-authors.